# Wheat microbiome bacteria can reduce virulence of a plant pathogenic fungus by altering histone acetylation

Yun Chen [1], Jing Wang[1], Nan Yang[1], Ziyue Wen[1], Xuepeng Sun[1], Yunrong Chai[2] & Zhonghua Ma [1]

Interactions between bacteria and fungi have great environmental, medical, and agricultural importance, but the molecular mechanisms are largely unknown. Here, we study the interactions between the bacterium *Pseudomonas piscium*, from the wheat head microbiome, and the plant pathogenic fungus *Fusarium graminearum*. We show that a compound secreted by the bacteria (phenazine-1-carboxamide) directly affects the activity of fungal protein FgGcn5, a histone acetyltransferase of the SAGA complex. This leads to deregulation of histone acetylation at H2BK11, H3K14, H3K18, and H3K27 in *F. graminearum*, as well as suppression of fungal growth, virulence, and mycotoxin biosynthesis. Therefore, an antagonistic bacterium can inhibit growth and virulence of a plant pathogenic fungus by manipulating fungal histone modification.

[1] State Key Laboratory of Rice Biology, Institute of Biotechnology, Zhejiang University, Hangzhou 310058, China. [2] Department of Biology, Northeastern University, Boston, MA 02115, USA. These authors contributed equally: Yun Chen, Jing Wang. Correspondence and requests for materials should be addressed to Y.C. (email: chenyun0927@zju.edu.cn) or to Z.M. (email: zhma@zju.edu.cn)

Plant physiology and performance strongly depend on the plant-associated microbiome under natural conditions[1]. Up to a few thousand bacteria, fungi, and protists, comprising a microbial community, colonize the roots and above-ground parts of plants[1–3]. Much research has been performed to evaluate how the environment and plant genes help to determine the structure of the plant microbiome. However, there is an increasing need for deciphering microbe–microbe interactions in the plant microbiome and functions of the interactions that drive the dynamic microbial community.

Bacteria and fungi are two major groups of the plant microbiome, and their interactions are critical in shaping the environmental microbial communities and exerting important effects on the fitness, colonization, or pathogenesis of the interacting partners[4]. Interactions and communications between bacteria and fungi can be achieved via antibiosis, signaling molecules, modulation of the physiochemical environment, chemotaxis, cooperative metabolism, protein secretion, or even gene transfer, leading to numerous biological effects that vary from antagonism to cooperation[5]. In agroecosystems, the most extensive studies on bacterial biocontrol agents (BCAs)–fungal communications have focused on antibiosis. For example, bacterial BCAs secrete lipopeptide antibiotics, phenazine derivatives, and other antifungal metabolites to directly inhibit *Fusarium graminearum*[6]. *Lysobacter enzymogenes* produces a heat-stable antifungal factor (HSAF) to inhibit ceramide synthase and degrade the fungal cell wall in *Aspergillus*[7]. On the fungal side, *Fusarium oxysporum* produces fusaric acid to suppress the production of 2,4-diacetylphloroglucinol by *Pseudomonas fluorescens* CHA0, and the biosynthesis of phenazine and a virulence-associated quorum sensing system in *Pseudomonas chlororaphis* PCL 1391[8,9]. In addition, beneficial bacteria degrade fungal virulence factors, produce volatile antifungal compounds, or induce plant systemic resistance against phytopathogenic fungi[10–12]. Although many BCAs existing in agricultural ecology have already been used for fungal disease control, the molecular mechanisms of antibiotics produced by BCAs remain to be elucidated.

Fusarium head blight (FHB) is predominately caused by *F. graminearum* (Fg) and is an economically devastating disease of small grain cereal crops[13]. Fg infection not only results in yield loss, but also contaminates grains with mycotoxins, such as deoxynivalenol (DON) and zearalenone, which pose a great threat to human and animal health[14]. Application of chemical fungicides remains the main approach to control FHB due to the lack of effective resistant wheat cultivars[15]. Unfortunately, fungicide-resistant Fg strains have been detected in the field after long-term intensive application of fungicides. Moreover, the application of several fungicides at sub-lethal concentrations triggers mycotoxin biosynthesis[16–18]. Biocontrol of FHB by BCAs represents an alternative approach and may be used as part of the integrated management of FHB and mycotoxin production.

In this study, we screen more than 12,000 culturable bacterial isolates from the wheat head microbiome, and obtain a potential BCA (*Pseudomonas piscium* ZJU60) with high antagonistic activity against FHB. We show that phenazine-1-carboxamide (PCN) secreted by ZJU60 directly targets the FgGcn5 protein, a histone acetyltransferase (HAT) of the Spt-Ada-Gcn5-acetyltransferase (SAGA) complex, subsequently resulting in deregulation of histone acetylation and suppression of fungal growth, mycotoxin biosynthesis, and virulence in Fg. In addition, ZJU60 forms biofilms on Fg hyphae, and PCN production is increased during bacterial–fungal interaction (BFI). Our study reveals a novel type of epigenetic regulation in antagonistic BFI.

## Results

**ZJU60 shows strong inhibitory activity against FHB.** In the rhizosphere, increasing evidence has shown that plants recruit protective bacteria, and enhance microbial activity to suppress pathogens[2,19]. To investigate whether the antagonistic bacterial community in wheat head is able to protect host plants during infection by Fg, bacterial communities associated with healthy and infected wheat heads were characterized by sequencing the V3–V4 region of the 16S rRNA gene. The sequences were grouped into 482 and 600 operational taxonomic units (OTUs) across 38 genera from healthy and infected wheat head samples, respectively. Sequencing data indicated that the relative abundance of bacterial genera in the microbiome was significantly altered after infection by Fg (Supplementary Table 1). At the genus level, *Pseudomonas* spp., *Paenibacillus* spp., *Sphingomonas* spp., and other commonly used plant biocontrol genera increased; in particular, the population of *Pseudomonas* spp. demonstrated a nearly 10-fold increase after Fg infection (Supplementary Table 1). Microbial community reassembly on the wheat head may include antagonistic bacteria in the microbiome to defend against infection by the fungus. However, due to the large number of species in this microbiome, it is difficult to study the functions involved in the interaction of the whole bacterial community with Fg on wheat head. Therefore, we focused on a simple culturable bacterium–Fg interaction system to investigate the roles of commensal bacteria in the suppression of FHB.

A total of 12,854 culturable bacterial isolates were obtained from wheat heads and examined for antagonistic activity to the Fg strain PH-1 (NRRL 31084) in vitro. Among them, 492 isolates (3.82% of the total) demonstrated various degrees of inhibitory activities against fungal growth (Supplementary Table 2). Notably, a bacterial isolate (termed ZJU60) obtained from infected wheat head produced green crystals on the top of its colony after incubation for 5 days (Fig. 1a). ZJU60 showed strong inhibitory activity against Fg during co-cultivation, producing a radius of inhibition zone >20 mm, and an inhibition zone >15 mm against several other fungal pathogens (Fig. 1b). To determine whether ZJU60 inhibited the growth of Fg in planta, we conducted biocontrol experiments both in a growth chamber and in the field (see Methods for details). Similar to phenamacril, a fungicide widely used to control FHB in China, treatment with ZJU60 by foliar spray almost completely suppressed Fg infection on wheat heads in a growth chamber assay (Fig. 1c). In field trials, ZJU60 consistently showed a biocontrol efficacy of 50–70% against FHB (Fig. 1d). Moreover, ZJU60 was able to significantly reduce DON production in field trials (Fig. 1e). These results indicate that ZJU60 is an effective BCA for the control of FHB.

Next, we applied multiple approaches to identify ZJU60. Results from gas chromatography cellular fatty acid analysis (GC-FA) and the high-throughput matrix-assisted laser desorption/ionization (MALDI) biotypersmart system showed that this strain had a similarity index of 0.745 (Supplementary Table 3) and score value of 2.081 (Fig. 1f) relative to *P. chlororaphis*. Since three possible species were suggested by CG-FA identification and a MALDI score value ranging from 2.000 to 2.299 only secures genus identification, we sequenced the complete genome of ZJU60 on a PacBio RSII. We obtained a single circularized chromosome of 6,818,002 bp in length with a GC content of 62.8% from ZJU60 (Fig. 1h). The phylogenetic relationship between ZJU60 and other sequenced *Pseudomonas* spp. was further analyzed by comparing ten housekeeping genes (*16SrRNA, aroE, dnaA, guaA, gyrB, mutL, ppsA, pyrC, recA,* and *rpoB*). ZJU60 was grouped together with *P. chlororaphis* strains and was most closely related to the biocontrol strain *P. chlororaphis* subsp. *piscium* PCL1391 (Fig. 1h), but genomic features of ZJU60 were distinct from other reference strains

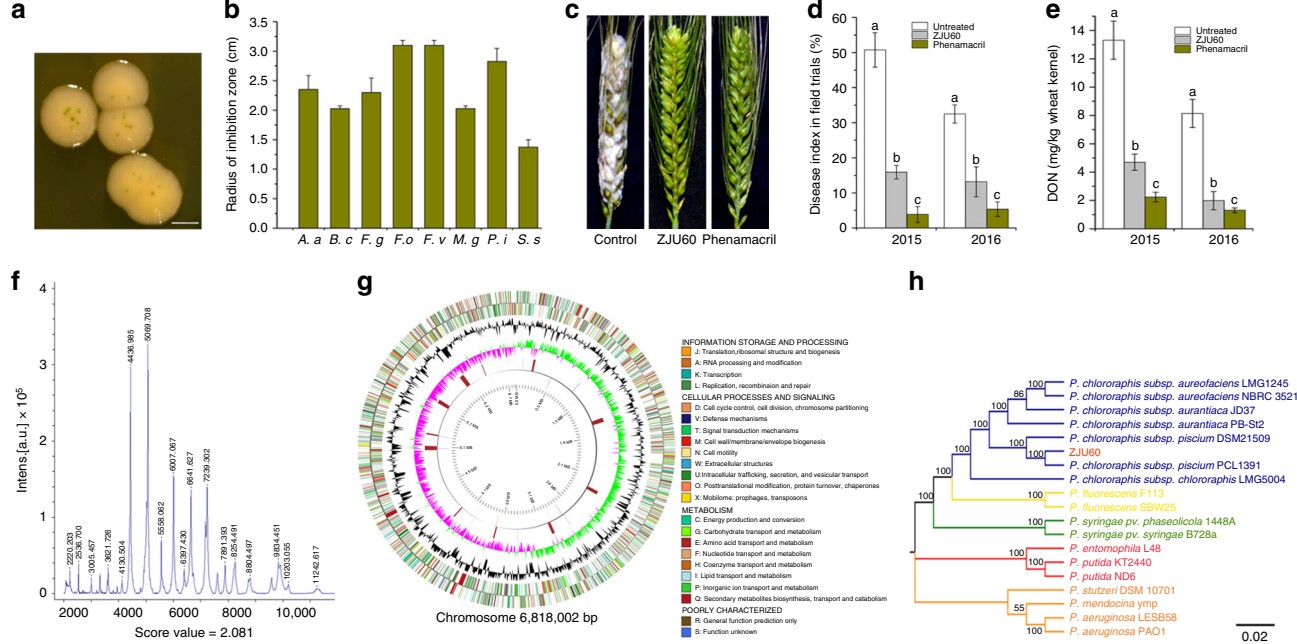

**Fig. 1** Isolation and identification of ZJU60 for FHB control. **a** Colony morphology of ZJU60 on King's B agar medium after 5 days of incubation at 30 °C. Bar = 1 mm. **b** Antagonistic activity of ZJU60 against eight important plant pathogenic fungi. The radius of inhibition zone (centimeter, cm) was measured for each co-culture of ZJU60 with *Alternaria alternata* (*A. a*), *Botrytis cinerea* (*B. c*), *Fusarium graminearum* (*F. g*), *Fusarium oxysporum* (*F. o*), *Fusarium verticillioides* (*F. v*), *Magnaporthe grisea* (*M. g*), *Penicillium italicum* (*P. i*), and *Sclerotinia sclerotiorum* (*S. s*). **c** Biocontrol efficacy of ZJU60 against FHB in a growth chamber. The fungicide phenamacril was used as a control. **d** The disease index of FHB after treatment with ZJU60 or fungicide in field trials conducted in 2015 and 2016. Water was used for the untreated control. **e** Efficiencies of ZJU60 and phenamacril in controlling DON production in field trials. Data presented are the mean ± s.d. ($n$ = 3). Bars followed by the same letter are not significantly different according to a Fisher's least significant difference (LSD) test at $P$ = 0.05. **f** ZJU60 profile generated by the high-throughput MALDI Biotypersmart system. **g** The circularized genome map of *Pseudomonas chlororaphis* subsp. *piscium* ZJU60. The outermost and second circles of all replicons indicate genes in forward and reverse orientations, and are color-coded by their role categories. The third circle shows G + C content in black and the fourth circle shows the G + C skew in green (+) and purple (−). The fifth circle shows the insertion sequences in red, putative prophage remnants in brown, and MITEs in blue. The scale is shown in the innermost circle. **h** A phylogenetic tree of ZJU60 with the other completely sequenced representatives of *Pseudomonas* spp. strains using ten housekeeping genes as phylogenetic markers. The significance of each branch is indicated by a bootstrap value calculated for 1000 subsets. Bar, 2 nucleotide substitutions per 1000 nucleotides

(Supplementary Table 4). Therefore, we identified ZJU60 as a new biocontrol strain of *P. chlororaphis* subsp. *piscium*.

**PCN is the major antifungal compound produced by ZJU60.** Bioinformatics analyses showed that the ZJU60 genome contained 11 putative gene clusters for the biosynthesis of secondary metabolites with potential antimicrobial activities (Fig. 2a), including phenazine, siderophores, and bacteriocin. Given that ZJU60 has the potent antifungal activity (Fig. 1), we searched for genes encoding potential antifungal compounds, and found that ZJU60 was predicted to secrete four different categories of antifungal compounds: hydrogen cyanide (HCN), phenazine, pyoverdine, and achromobactin, which are distinctly different from those of other strains of *P. chlororaphis* used as BCAs (Supplementary Table 5). To further identify antifungal compounds produced by ZJU60, deletion mutants of the individual operons involved in the biosynthesis of each potential antifungal compound were constructed, and the antifungal activity of each mutant was examined. As shown in Fig. 2b, c, disruption of the *phzA-H* operon, which is responsible for the biosynthesis of phenazine compounds, resulted in a complete loss of antifungal activity both in vitro and in planta. In contrast, the antifungal activity of the mutants that disrupted biosynthetic operons for HCN, pyoverdine or achromobactin, was not significantly different from that of the wild-type ZJU60 (Supplementary Fig. 1, Fig. 2b). These data suggest that the phenazine compounds

produced by ZJU60 may be critical for its antifungal activity against Fg.

The *phzA-H* operon is known to be responsible for the biosynthesis of PCN from the precursor chorismic acid[20] (Fig. 2d). Furthermore, the production of phenazine compounds by ZJU60 was confirmed by extraction and identification using liquid chromatographic-mass spectrometry (LC-MS). The major phenazine compound was shown to be PCN (84%), followed by the intermediate product phenazine-1-carboxylic acid (PCA, 16%) (Fig. 2e). The growth inhibitory activities of these two phenazine compounds toward Fg were determined in vitro. As shown in Fig. 2f, g, PCN was more potent than PCA against Fg growth. Taken together, our results indicate that PCN is the major antifungal compound produced by ZJU60.

To further confirm the antifungal activity of PCN, the morphology of Fg hyphae treated with PCN was observed by scanning electron microscopy (SEM) and transmission electron microscopy (TEM). These experiments showed that Fg hyphae were dramatically misshapen after treatment with 25 μg ml⁻¹ PCN (the concentration producing 90% inhibition, $EC_{90}$ = 25 μg ml⁻¹) for 6 h (Supplementary Fig. 2). Taken together, our results indicate that PCN is the major antifungal compound produced by ZJU60.

**PCN mediates the BFI and biofilm formation of ZJU60.** Interestingly, ZJU60 produced more PCN (green crystals[21])

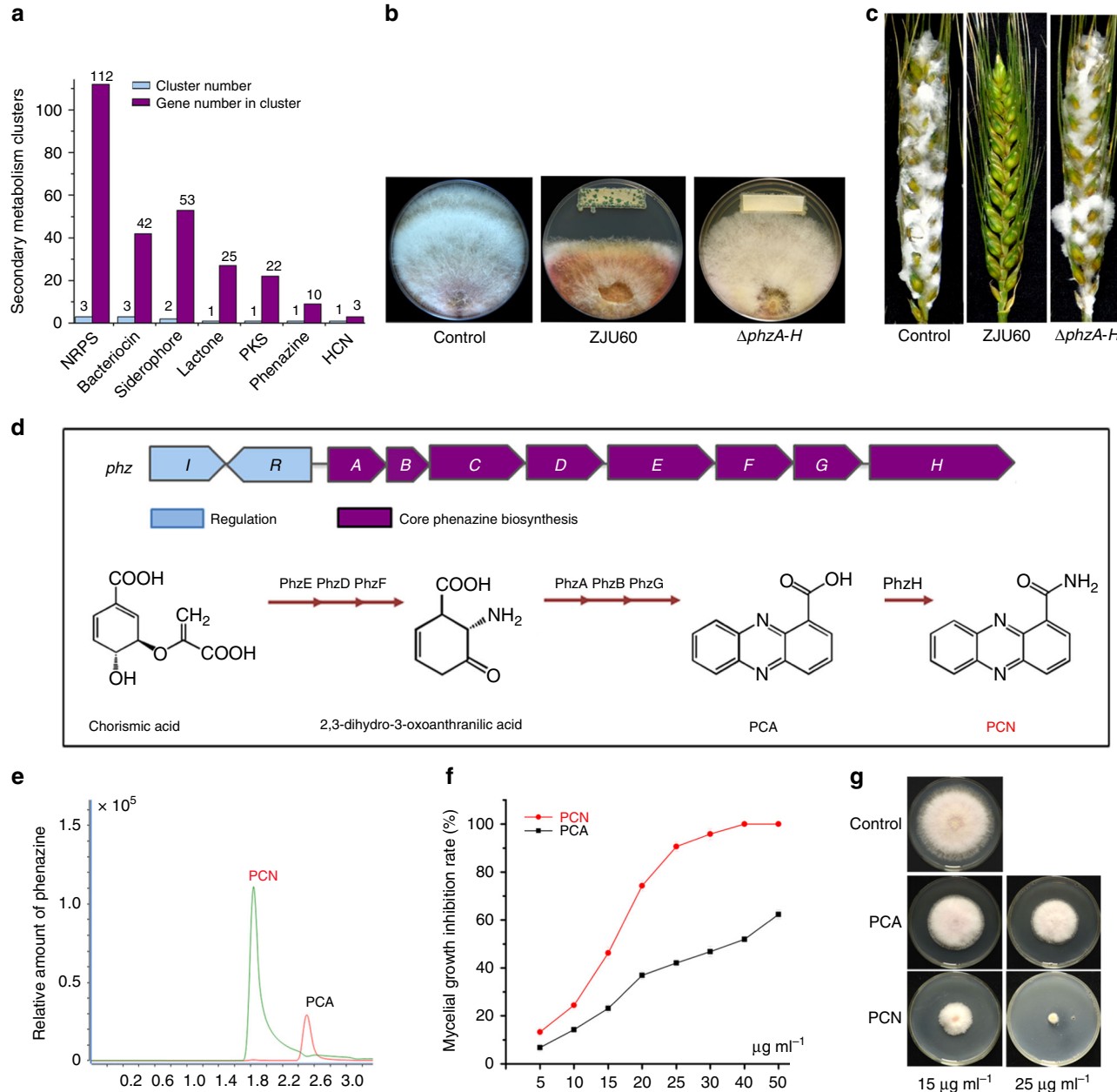

**Fig. 2** PCN is the major antifungal compound produced by ZJU60. **a** Predicted secondary metabolite gene clusters in the ZJU60 genome using anti-SMASH. **b** Disruption of the PCN biosynthesis cluster *phzA-H* completely abolished the antagonistic activity of ZJU60 towards Fg in a co-culture assay. **c** Effects of ZJU60 and Δ*phzA-H* on the growth inhibition of Fg *in planta*. **d** Predicted PCN biosynthesis pathway in ZJU60. **e** Identification of phenazine compounds and their relative amounts in the supernatant of ZJU60 by LC-MS. **f** Mycelia growth inhibition curve by PCN or PCA at various concentrations. **g** Growth inhibitory activity exerted by PCN or PCA on Fg at final concentration of 15 and 25 μg ml$^{-1}$

during co-culture with Fg on a Warkingsman's agar (WA) plate than ZJU60 alone (Fig. 3a). In addition, Fg mycelia were heavily decorated by PCN during co-culture with ZJU60 in WA liquid medium (Fig. 3b). This observation implied that PCN biosynthesis may be stimulated in ZJU60 during co-culture with Fg. Thus, we extracted and quantified PCN production in ZJU60 cultured alone and ZJU60 co-cultured with Fg using LC-MS. As shown in Fig. 3c, relative PCN production was approximately 50% greater in the co-culture than that in ZJU60 pure culture. These data suggested that physical interactions between ZJU60 and Fg stimulated PCN production.

The interaction of ZJU60 and Fg was further examined by SEM. After 5 days of co-culture in WA liquid medium, a large number of ZJU60 cells formed extensive biofilm-like structures

on the surface of fungal mycelia and caused shriveled mycelia (Fig. 3d). However, very few bacterial cells were associated with fungal mycelia during the co-culture of Fg with the PCN biosynthesis defective mutant Δ*phzA-H* (Fig. 3e). In addition, the fungal mycelia in the co-culture (Δ*phzA-H* vs Fg) were still plump (Fig. 3e) and have a morphology similar to that of Fg mycelia grown alone (Supplementary Fig. 3a). To test whether the reduced bacterial colonization of Δ*phzA-H* on the surface of fungal mycelia was caused by the growth defect of the PCN biosynthesis mutant Δ*phzA-H*, the population of mycelium-associated bacteria was counted during the co-culture of Δ*phzA-H* vs Fg and the co-culture ZJU60 vs Fg by plating on WA plates. As shown in Fig. 3f, the total number of bacterial cells including both the mycelium-associated and planktonic cells were not

significantly different between the two co-cultures. However, the number of just mycelium-associated bacterial cells in the ZJU60 vs Fg co-culture was significantly higher (20-fold) than that in the Δ*phzA-H* vs Fg co-culture (Fig. 3g). These results indicated that disruption of PCN biosynthesis impaired colonization of this bacterium on the fungal mycelium.

Since biofilm plays an important role in bacterial colonization on biotic and abiotic surfaces[22,23], the biofilm formation ability by

ZJU60 and Δ*phzA-H* was further evaluated in vitro. In yeast extract peptone dextrose (YEPD) medium, ZJU60 cells formed thin floating pellicle biofilms[24] at the air–liquid interface (Fig. 3h, left-hand panels). In addition, cells also formed submerged biofilms in WA and YEPD media, especially WA medium (Fig. 3h, top panels). However, Δ*phzA-H* was deficient in biofilm formation and formed thin and fragile pellicles in YEPD (Fig. 3h, bottom panels). The submerged biofilms associated with the

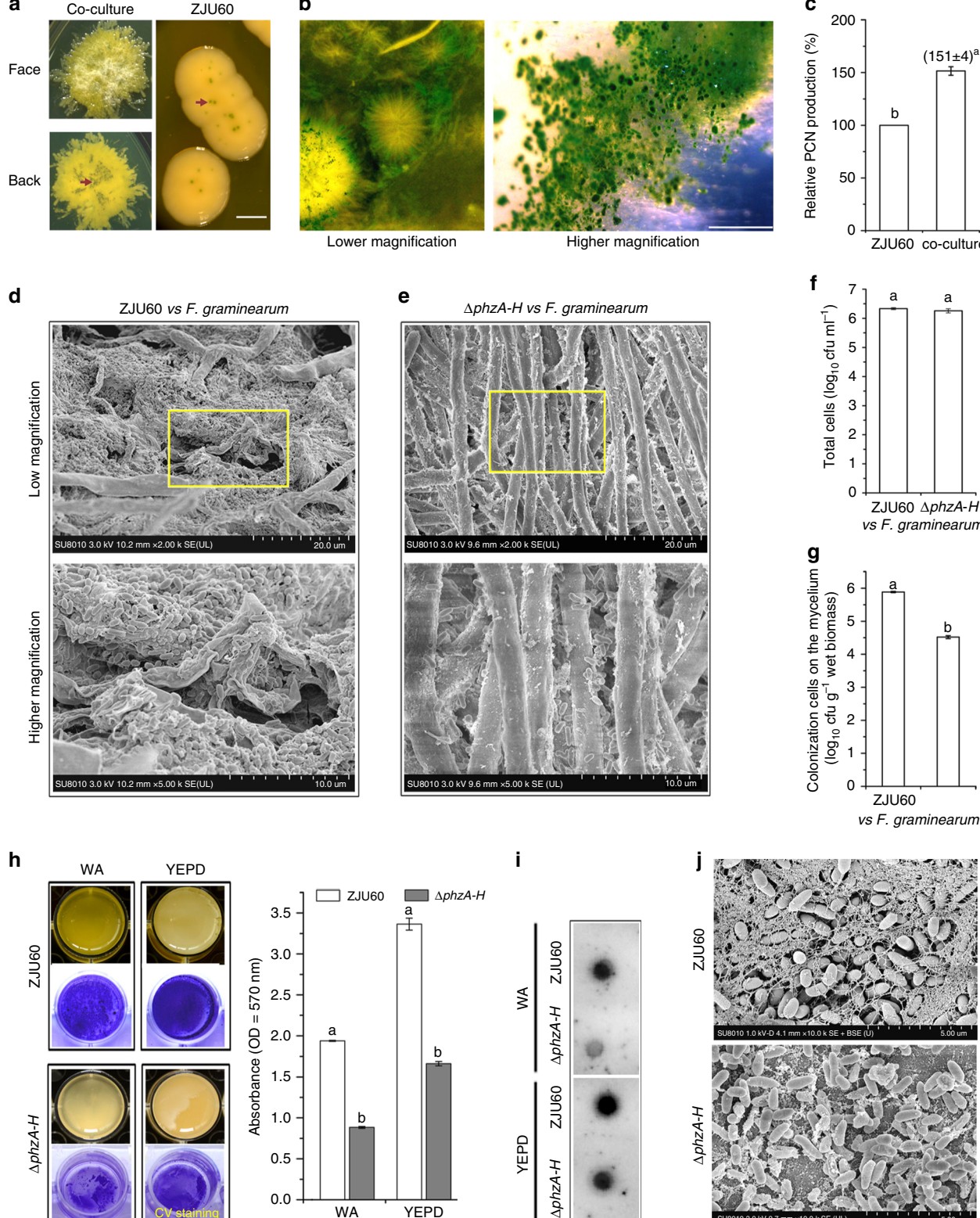

bottom surface of the wells of the microtiter plate were visualized by crystal violet (CV) staining[25]. Upon staining, the biomass of $\Delta phzA$-$H$ adhering to the wells was visibly thinner and sparser than that of ZJU60 (Fig. 3h, right-hand panel). Exopolysaccharides are a key biofilm matrix component in many bacteria because they contribute to the overall biofilm architecture[24]. Genome analysis of the potential gene clusters responsible for exopolysaccharide biosynthesis showed that ZJU60 contained a polysaccharide locus (PsI) homologous cluster (Supplementary Fig. 4). PsI encodes mannose-rich fiber-like structural polysaccharides of biofilm matrix in *Pseudomonas aeruginosa*[26,27]. Therefore, PsI polysaccharide production by ZJU60 and $\Delta phzA$-$H$ was also quantified by immunoblotting with anti-PsI serum[27]. Consistent with the CV staining results, PsI production by the mutant was dramatically lower than that of the wild-type ZJU60 in both media tested (Fig. 3i). Under SEM, cells were heavily cross-linked by a self-produced extracellular fiber-like matrix in ZJU60 biofilms, while no visible matrix fibers were observed in the biofilms of $\Delta phzA$-$H$ under the same conditions (Fig. 3j). In addition, ZJU60 also formed a biofilm-like macro-colony with fibers on the surface of wheat tissues (Supplementary Fig. 3b). Collectively, our results suggested that PCN may play an important role in biofilm formation in ZJU60 and that biofilms physically mediate the interactions between ZJU60 and Fg.

**SAGA complex is critical for PCN sensitivity in Fg.** To gain insight into the mode of action exerted by PCN on fungal growth inhibition, we first conducted chemical genomic analyses in *Saccharomyces cerevisiae*[28]. We treated *S. cerevisiae* with 10 μg ml$^{-1}$ PCN and recovered a total of 90 mutants with potentially altered sensitivity to PCN were recovered by this approach (Supplementary Fig. 5a; Supplementary data 1). To confirm these results, we tested the PCN sensitivity of the 90 individual mutants as well as that of 345 other mutants defective in their interacting proteins (Supplementary data 2). Among them, four mutants (*sod1*, *lsm7*, *npt1*, and *elm1*, Supplementary Table 6) dramatically changed their sensitivity to PCN (Supplementary Fig. 5b-c). We thus constructed mutants of those homologous genes in Fg. Surprisingly, none of the single or double homologous gene mutants in Fg showed altered sensitivity to PCN (Supplementary Fig. 5d-e). In addition, phenazines have been proposed to inhibit fungal growth by inducing the production of reactive oxygen species (ROS), specifically superoxide free radicals $O_2^{\cdot-}$[29,30]. However, we found that superoxide dismutase (SOD) mutants of Fg ($\Delta Fgsod1$, $\Delta Fgsod1/sod2$, and $\Delta Fgmnsod1/mnsod2$) did not show elevated sensitivity to PCN (Supplementary Fig. 5d-e) although these mutants were hypersensitive to menadione, which is known to generate intracellular $O_2^{\cdot-}$ (Supplementary Fig. 5f). Thus, it is conceivable that the growth inhibition by PCN in *S. cerevisiae* and Fg is achieved through different molecular mechanisms.

To further investigate the antifungal mechanism exerted by PCN on Fg, we then tested PCN sensitivity of 1100 Fg deletion mutants available in our laboratory. Proteins encoded by those mutated genes are known to be involved in various biological functions, including putative phosphatases (71 genes)[31], protein kinases (96 genes)[32], ATP-binding cassettes (60 genes), transcription factors (105 genes), and epigenetic modification related proteins (72 genes) (gene IDs are shown in Supplementary data 3). Surprisingly, of the 1100 deletion mutants screened, only one mutant ($\Delta FGSG\_06291$) showed significantly increased sensitivity to PCN (Fig. 4a), and the mutant gene was designated as *FgPCN*. To verify that the PCN-sensitive phenotype of $\Delta FgPCN$ was not caused by reduced mycelial growth, we extended the incubation time to 5 days until the mycelia of $\Delta FgPCN$ without PCN treatment reached the edge of the plate. As shown in Supplementary Fig. 6a, $\Delta FgPCN$ was still unable to grow on CM plates supplemented with 15 μg ml$^{-1}$ PCN. Moreover, a complementation strain (designated $\Delta FgPCN$-$C$) fully rescued the PCN sensitivity of the mutant (Supplementary Fig. 6b). Bioinformatics analysis showed that *FgPCN* was homologous to *SPT7* in *S. cerevisiae*, encoding a transcriptional activator subunit in the SAGA transcriptional regulatory complex[33]. Intriguingly, the mutant of $\Delta SPT7$ in *S. cerevisiae* did not show altered sensitivity to PCN (Supplementary Fig. 7a). Sequence alignment analyses indicated that only 27% of the amino acid sequence is identical between FgPcn and Spt7. Further phylogenetic analysis indicated that fungal Spt7 proteins had two different evolutionary paths and were divided into yeast and filamentous fungal groups (Supplementary Fig. 7b). These results implied that the divergence in amino acid sequences of Spt7 orthologs may be responsible for the different sensitivities to PCN in yeast and filamentous fungi. To further test this hypothesis, we constructed the deletion mutants of the Spt7 ortholog ($BC1G\_13923$) in the filamentous fungus *Botrytis cinerea* and evaluated its sensitivity to PCN. As shown in Supplementary Fig. 7c, the mutant $\Delta BcSPT7$ of *B. cinerea* demonstrated greater sensitivity to PCN than the parent strain. These results support the idea that the *FgPCN* gene is associated with PCN sensitivity in Fg.

SAGA, the largest HAT complex, consists of 19 core subunits; and in *S. cerevisiae*, the subunit Gcn5 serves as the acetyltransferase[34]. We therefore sought to determine whether the gene disruption of other SAGA subunits may also affect the PCN sensitivity in Fg. Genes encoding five other key SAGA subunits (FgAda3, FgGcn5, FgSgf73, FgSpt8, and FgSpt3) were selected for targeted gene deletion (Supplementary Table 7), and the resulting mutants were examined for PCN sensitivity. As shown in Fig. 4a, b, $\Delta FgADA3$, $\Delta FgGCN5$ and to a lesser extent $\Delta FgSGF73$, $\Delta FgSPT8$, and $\Delta FgSPT3$, all demonstrated dramatically increased to PCN (Fig. 4a, b), similar to that observed from $\Delta FgPCN$. Moreover, the corresponding complementation strains all rescued the PCN sensitivity of the mutants (Supplementary Fig. 6b). Consistently, these mutants also showed increased sensitivity to

**Fig. 3** PCN production is enhanced in the ZJU60–Fg interaction. **a** Colony morphology of co-cultured ZJU60 and Fg on Warkingsman's agar (WA) agar medium after 5 days of incubation. The green pigments (PCN) are indicated by arrows. Bar = 1 mm **b** Features of fungal mycelium after 5 days of co-culture with ZJU60 in WA liquid medium. The details of the green crystals are enlarged (right-hand panel), bar = 1 mm. **c** Relative PCN production of ZJU60 and a ZJU60–Fg co-culture in WA liquid medium. **d** ZJU60 biofilm features on the mycelium of Fg during co-culture. **e** Bacterial colonization features of the $\Delta phzA$-$H$ mutant on the mycelium of Fg during co-culture. **f** The total number of planktonic and mycelium-associated bacterial cells during co-culture for two bacterial–fungal combinations after 5 days of incubation. **g** The population of fungal mycelium-associated bacterial cells after 5 days of co-culture, excluding the planktonic cells. **h** Biofilm formation by ZJU60 and $\Delta phzA$-$H$ in WA and yeast extract peptone dextrose (YEPD) media. The submerged biofilm was stained with crystal violet (CV, bottom panel). The average amount of CV retained in the bacterial biomass was quantified (right-hand panels) from three independent experiments consisting of four internal replicates. **i** PsI polysaccharides in the biofilm matrix of ZJU60 and $\Delta phzA$-$H$ formed in YEPD and WA media were detected by immunoblotting with anti-Psl serum. **j** Morphological features of the individual biofilm of ZJU60 and $\Delta phzA$-$H$ were visualized by SEM. Data presented are the mean ± s.d. ($n = 3$). Bars followed by the same letter are not significantly different according to Student's *t* test at $P = 0.05$

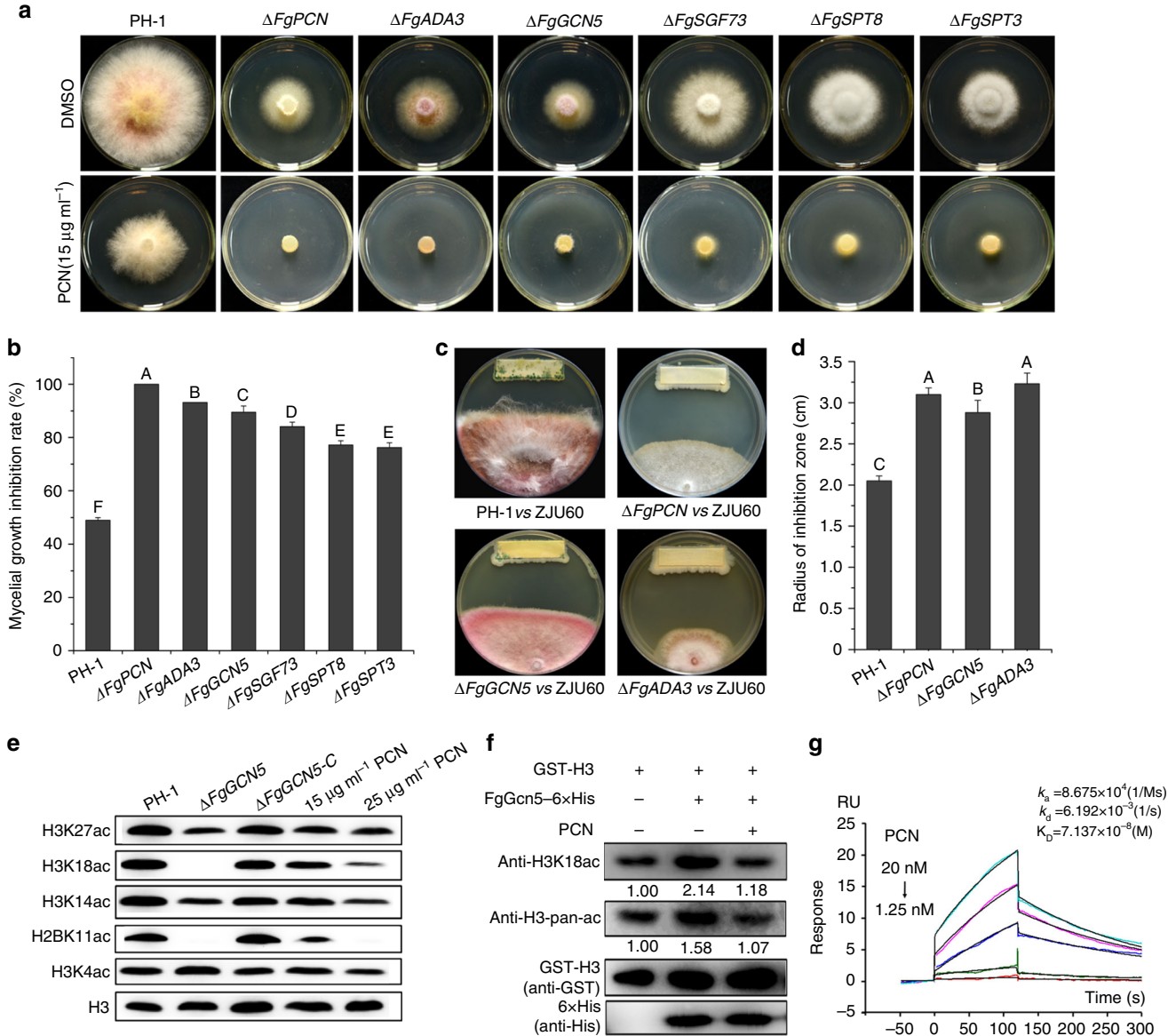

**Fig. 4** PCN targets the acetyltransferase FgGcn5 of the SAGA complex in Fg. **a** PCN sensitivity of deletion mutants for the six key components of the SAGA complex. **b** The mycelial growth inhibition rate was calculated after 3 days of incubation. **c** Antagonistic activity of ZJU60 towards wild-type PH-1 and ΔFgPCN, ΔFgADA3, and ΔFgGCN5 mutants. **d** The radius of the inhibition zone was measured after 5 days of incubation. Data presented are the mean ± s.d. (n = 3). The same letter on the bars for each column indicates no significant difference according to a LSD test at P = 0.01. **e** Effects of FgGcn5 deletion and PCN treatment on histone acetylation. Several antibodies against specific lysine (K) acetylation (ac) sites on histones were used to detect histone acetylation in western blot analyses. Hybridization with an anti-H3 C-terminus antibody served as a loading control. **f** Inhibition of activity against the acetyltransferase activity of FgGcn5 on H3 and H3K18 in vitro. Purified recombinant GST-H3 was incubated with FgGcn5-6xHis in HAT buffer with or without PCN for 45 min at 37 °C and then immunoblotted with the indicated antibodies. The relative intensities of H3K18ac and H3-pan-ac bands were quantified with ImageJ, and the fold-change relative to that of without FgGcn5 control is indicated below the bands. The bands in the control reaction (without 6xHis-FgGcn5) were set as 1.00. **g** SPR analysis of PCN binding to FgGcn5. SPR kinetic sensorgrams (colored lines) and fitting curves (black lines) for PCN binding to FgGcn5 at different concentrations (from top to bottom: 20, 10, 5, 2.5, 1.25 nM). The responses were reference-subtracted, solvent-corrected, and blank-deducted

ZJU60 (Fig. 4c, d). In addition, SEM examination revealed that ΔFgPCN, ΔFgADA3, and ΔFgGCN5 mutants grew slowly with tortuous mycelia without PCN treatment, similar to the wild-type mycelia treated with PCN (Supplementary Fig. 6c). Taken together, these data indicate that the SAGA complex plays a role in PCN sensitivity in Fg.

**PCN specifically targets the HAT activity of FgGcn5 in SAGA.** SAGA is responsible for histone acetylation, which subsequently regulates the transcription of RNA polymerase II-dependent genes. To gain further insight into the possible mechanism underlying PCN in SAGA-mediating histone modification in Fg, several putative acetylation sites at histones (see Methods for details) were analyzed at a global level using immunoblotting analysis. Acetylation of Histone 2BK11 (H2BK11) and Histone 3K18 (H3K18) was undetectable in the acetyltransferase mutant ΔFgGCN5 (Fig. 4e). Similarly, the acetylation level at H3K18 was undetectable in wild-type cells treated with 25 μg ml⁻¹ PCN. Moreover, 25 μg ml⁻¹ PCN dramatically inhibited the acetylation of H2BK11 (Fig. 4e). In addition, the acetylation levels at H3K14

and H3K27 were partially reduced in both $\Delta FgGCN5$ and the wild-type mycelia treated with PCN. These results suggested that PCN may inhibit the HAT activity of FgGcn5, and subsequently affect global histone acetylation. To verify this possibility, we evaluated the effect of PCN on the acetyltransferase activity of FgGcn5 on histone 3 (H3) in vitro. The GST-H3 and FgGcn5-6×His were expressed in *Escherichia coli* and purified with GST agarose and a Ni-NTA column equipped on an AKTA purification system, respectively (details in Supplementary Methods). As shown in Fig. 4f, although GST-H3 was partially acetylated when expressed in *E. coli*, FgGcn5 was still able to stimulate higher levels of H3 global acetylation and H3K18 acetylation (H3K18ac) (the tested specific catalyzing site in H3 by FgGcn5). Importantly, the global acetylation levels of H3 (H3-pan-ac) and H3K18ac notably decreased to basal levels upon the addition of PCN, demonstrating that PCN was able to inhibit the HAT activity of FgGcn5 in vitro. Next, we examined whether PCN could directly bind to FgGcn5 using a direct surface plasmon resonance (SPR) biosensor. SPR analysis revealed that the target affinity [KD (equilibrium dissociation constant) value] of PCN binding to FgGcn5 was 71.37 nM, indicating a strong binding affinity of PCN for FgGcn5 (Fig. 4g). Furthermore, the binding positions of PCN to the HAT of FgGcn5 were predicted using the multiple ligand simultaneous docking (MLSD) program, and two PCN molecules likely bind within the FgGcn5 pocket, creating a stable PCN–FgGcn5 complex (Supplementary Fig. 8, Supplementary movie 1-3). Together, these data indicate that PCN likely directly inhibits the HAT activity of FgGcn5.

To determine whether PCN also affects the activities of other acetyltransferases in this fungus, we searched all putative genes encoding either HAT or N-acetyltransferase in the *F. graminearum* genome, and constructed deletion mutants for each of these genes (a total of 31). Among the resulting mutants, only $\Delta FgGCN5$ showed significantly increased sensitivity to PCN (Supplementary Table 8), which indicated that PCN targeted specifically the HAT FgGcn5 in the SAGA complex, but did not affect the activity of other acetyltransferases in Fg.

We next tested whether the SAGA complex is also responsible for the sensitivity of Fg to other antifungal drugs. Three widely used fungicides, fludioxonil, phenamacril, and triadimefon targeting MAPK signal transduction, motor protein myosin I, and sterol biosynthesis, respectively, were selected. As shown in Supplementary Fig. 9, deletion mutants of key SAGA components did not alter their sensitivity to these three antifungal drugs, indicating that the SAGA complex may be specifically associated with PCN sensitivity in Fg.

**SAGA complex regulates growth and virulence of Fg**. To analyze the influence of the Fg SAGA complex on fungal growth and differentiation, six deletion mutants of the SAGA components, $\Delta FgPCN$, $\Delta FgADA3$, $\Delta FgGCN5$, $\Delta FgSGF73$, $\Delta FgSPT8$, and $\Delta FgSPT3$, were characterized. All tested mutants were impaired for hyphal growth. Moreover, all the mutants formed fewer velvet aerial hyphae with compact colonies on the agar surface (Fig. 4 and Supplementary Fig. 6). In addition, none of the mutants produced conidia after incubation at 25 °C for 4 days in a shaker (180 rpm). Complementation of the deletion strains with the corresponding wild-type gene under the control of its native promoter fully restored the defects in hyphal growth and conidiation. Because ZJU60 targeting the SAGA complex reduced the disease severity of FHB (Fig. 1), the role of the SAGA complex in pathogenicity was also determined. All strains were assayed by point inoculation of wheat heads at anthesis using mycelia since the mutants were unable to produce conidia. Fifteen days after inoculation, the wild-type and complementation strains were able

to infect tissue and cause typical FHB symptoms. However, all six mutants were non-pathogenic on wheat (Fig. 5a). DON, a critical virulence factor, is synthesized in the specific cellular compartment known as the toxisome in Fg, which contains trichothecene biosynthesis enzymes encoded by *TRI* genes[17]. To investigate whether the SAGA complex regulates DON biosynthesis, we examined toxisome formation by measuring Tri1-GFP[17], a key DON biosynthetic enzyme, and measured final DON production in the $\Delta FgPCN$, $\Delta FgADA3$, and $\Delta FgGCN5$ mutants in the DON-inducing medium TBI. As shown in Fig. 5b, the wild-type strain formed green spherical and crescent shaped toxisomes under DON-inducing conditions. In contrast, the green fluorescent signals of toxisomes were not observed in any of the individual mutants. The expression of Tri1-GFP in all tested strains was further verified by immunoblot assays using an anti-GFP antibody. In agreement with the microscopic observation, there was no detectable Tri1-GFP in the total protein samples of the mutants (Fig. 5c). Consistently, all tested mutants were unable to produce detectable levels of DON (Fig. 5d). Furthermore, these mutants clearly abolished toxisome formation in the hyphae of Fg inoculated on wheat leaves (Fig. 5e). Next, we determined the impact of PCN on DON biosynthesis; treatment with the fungicide carbendizm was included as a control to exclude the impact of a growth defect. As in the SAGA component mutants, PCN at either $EC_{50}$ or $EC_{90}$ significantly suppressed Tri1-GFP expression, disrupted toxisome formation, and reduced DON production in the wild-type strain, while carbendazim enhanced DON biosynthesis as reported previously[17] (Fig. 5b–d).

The deletion mutants of the SAGA components exhibited dramatically decreased virulence on wheat heads, even the inoculated spikelets; this phenotype was more severe than the virulence reduction exhibited by DON biosynthetic gene mutants, such as $\Delta tri6$ and $\Delta tri10$[35]. These finding indicated that loss of virulence was not entirely attributable to DON reduction in the SAGA mutants. Next, we found that the SAGA complex was also critical for the formation of fungal penetration structures. As shown in Supplementary Fig. 10a, the wild-type and complemented strains but not the mutants were able to penetrate cellophane sheets in the cellophane penetration assay. In addition, PCN supplemented in agar plates dramatically suppressed the ability of the wild-type to penetrate cellophane sheets (Supplementary Fig. 10b). More importantly, examination by SEM showed that the hyphae of the wild-type and corresponding complemented strains formed typical penetration structures on wheat spikelets, but such penetration structures were not observed for the mutants (Fig. 5f and Supplementary Fig. 10c). Using qRT-PCR, we found that the expression levels of penetration related genes, including genes in the FgGpmk1 mitogen-activated protein kinase cascade[36] and genes encoding plant cell wall degrading enzymes for infection[36,37] (supporting information data 4), were significantly decreased in the $\Delta FgGCN5$ mutant and the wild-type treated with PCN (Supplementary Fig. 11). Thus, these results suggested that the SAGA complex regulated the transcription of penetration related genes and was essential for infection structure formation on wheat tissues. Collectively, the abolishment of DON biosynthesis and penetration defects were mainly responsible for the non-pathogenic characteristics of the SAGA mutants on wheat, where growth defects of mutants might also play a minor role.

## Discussion

A considerable number of bacterial strains, especially *Pseudomonas* spp., have been used to directly control many diseases affecting several agricultural crops as BCAs. The most commonly produced phenazines by *Pseudomonas* spp. include PCA, PCN,

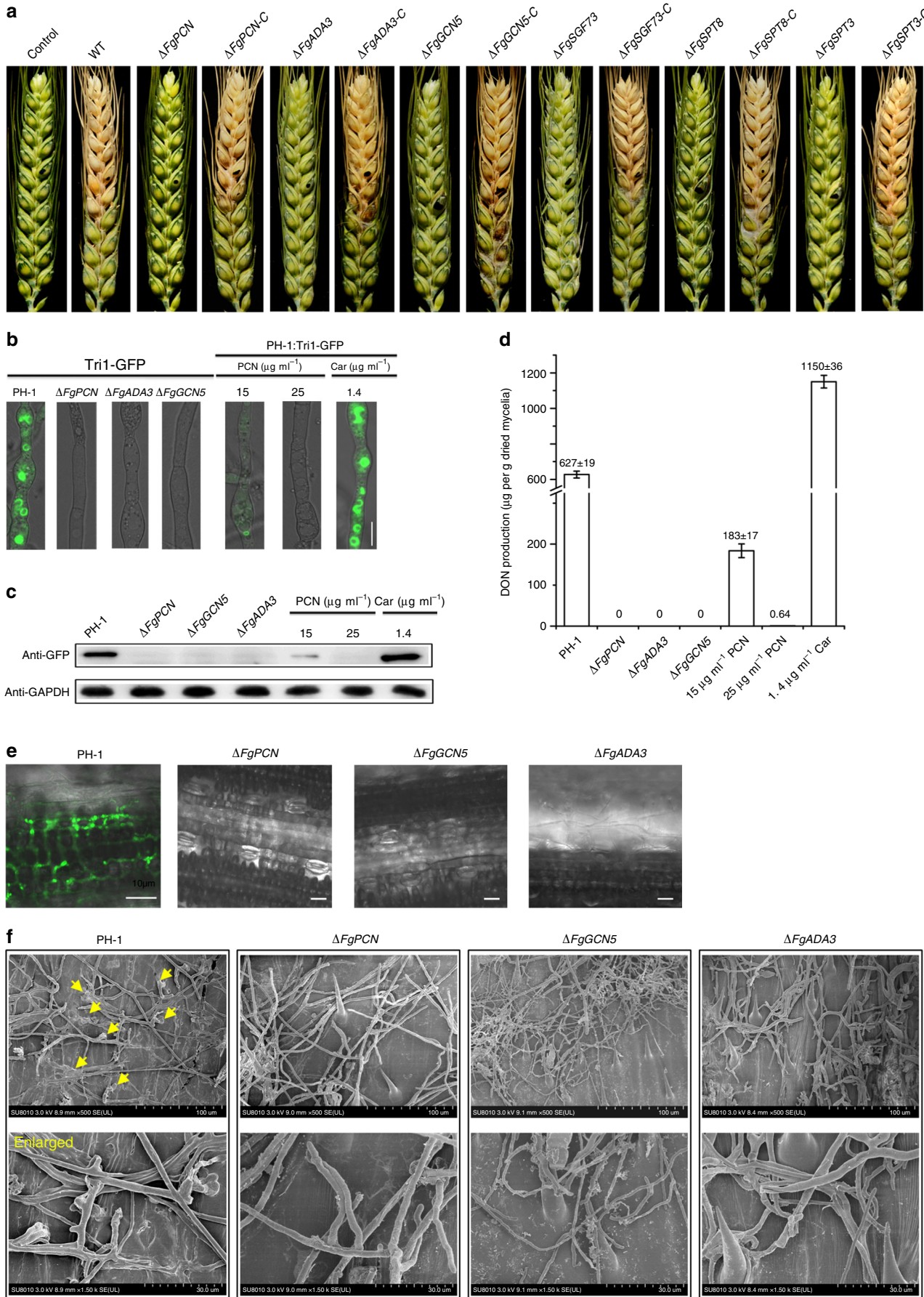

hydroxyphenazines, and pyocyanin, which are essential for the biocontrol efficiency of bacterial BCAs against fungal diseases[38,39]. Although phenazine-producing bacteria for the biocontrol of fungal diseases have been studied extensively over the past 3 decades, the exact mechanisms underlying the actions of phenazines in antifungal interactions remain unclear[38]. Multiple modes of action of phenazines against bacteria or fungi have been proposed, including the induction of ROS, DNA interaction, and inhibition of vacuolar ATPase activities and topoisomerase[29,40]. In this study, we isolated the *P. piscium* ZJU60 from infected wheat head and found that ZJU60 secreted PCN to inhibit Fg growth (Figs. 1 and 2). Redox stress was likely involved in the antifungal activity of PCN against yeast (Supplementary Fig. 5b), while Fg SOD mutants did not show altered sensitivity to PCN (Supplementary Figs. 5d-e). By screening a gene deletion mutant library and performing genetic and biochemical characterization, we discovered that PCN directly targets the FgGcn5 protein and subsequently inhibits the HAT activity of the SAGA complex in Fg (Fig. 4, Supplementary Figs. 6 and 8). Consistently, genetic disruptions of the SAGA complex significantly reduced hyphal growth, DON production and attenuated virulence of Fg (Fig. 5). Moreover, PCN was critical for ZJU60 biofilm formation on the hyphae of Fg during BFI (Fig. 3). Our results for the first time show that PCN inhibits fungal growth and virulence through the suppression of HAT activity of the SAGA complex (Fig. 6).

A growing body of evidence indicates that histone acetylation is an important mechanism mediating inter-species or even inter-kingdom interactions. To date, several pathogens have been reported to affect the chromatin structure and transcriptional program of host cells by altering histone acetylation[41]. For example, *Listeria monocytogenes* induces deacetylation of H3K18 mediated by the host deacetylase SIRT2 to promote infection in mammal cells[42]. The oomycetes of *Phytophthora sojae* secrete the effector PsAvh23 to manipulate host histone acetylation and subsequently suppress plant immunity[43]. More recently, histone acetylation was found to be involved in the bacteria–fungi interaction. *Streptomyces rapamycinicus* induces the biosynthesis of antimicrobial compounds in *Aspergillus nidulans* by activating histone acetylation in *A. nidulans*[44], although the bacterial secreted effector(s) for histone modification was not identified. In this study, we discovered that the bacterial BCA *P. piscium* ZJU60 secreted PCN to inhibit Fg histone acetylation by targeting FgGcn5, and subsequently repressing fungal growth (Fig. 4 and Supplementary Fig. 6). To the best our knowledge, this is the first report describing a bacterium able to inhibit fungal growth by manipulating fungal histone acetylation during the BFI.

Bacterial biofilms are structured microbial communities embedded in a self-produced extracellular polymer matrix. In most natural environments, bacteria exist predominantly on biotic and abiotic surfaces as biofilms rather than as planktonic cells[45]. In our previous studies, we found that *Bacillus* spp. BCA formed biofilms on plant root surfaces, and enhanced their biocontrol efficacies[45,46]. Recent studies have shown that biofilms are also important for interactions between bacteria and fungi in rhizosphere soil and clinical contexts. For instance, *Salmonella enterica* cells were shown to form biofilms on the hyphae of *Aspergillus niger* in a maize microbiome[47]. *P. aeruginosa*, an opportunistic bacterial pathogen, forms a dense biofilm on hyphae of the human fungal pathogen *Candida albicans*, and kills the fungus[21,48]. Arbuscular mycorrhizal fungi stimulate mycorrhizal helper bacteria to form biofilms on fungal hyphae[49]. Therefore, molecular communication in bacterial–fungal mixed biofilms has been a recent hot topic. To date, several extracellular signaling molecules, such as autoinducer-2, farnesol, N-acetyl-D-glucosamine, and homoserine lactones[50] have been described to play important roles during the construction of bacterial biofilms on fungal hyphae. Recently, Chen and colleagues reported that ethanol secreted by *C. albicans* was able to stimulate *P. aeruginosa* WspR-controlled biofilm formation and PCN production on fungal hyphae. Moreover, phenazines enhanced ethanol production by *C. albicans* and formed a feedback cycle to drive this BFI[21]. In this study, we also observed that ZJU60 formed a biofilm on the surface of fungal mycelia and produced more PCN when co-cultured with Fg (Fig. 3a–c). Previous studies have shown that the production of phenazines is tightly linked with the process of biofilm formation in phenazine producing *Pseudomonas* spp. Mutations in regulatory and structural phenazine genes caused deficiency in biofilm formation of *P. chlororaphis* and *P. aeruginosa*[51,52]. Phenazines could contribute to biofilm development through bacterial surface migration[53], ferrous iron acquisition[52], promoting extracellular DNA (eDNA) release[54], or serving as signals to trigger the expression of other important factors in biofilm development[51]. Consistence with that, PCN is critical for ZJU60 biofilm formation on the hyphae of Fg during BFI. These results extend our knowledge regarding the functions of antibiotics during BFI.

Although the SAGA complex is conserved from yeast to humans, the composition and function of this complex in filamentous fungi are largely unknown. In yeast, the SAGA complex assembles into a stable complex with an overall mass of ~1.8 MDa[34]. The other subunits of SAGA confer additional functionalities to the complex and are functionally organized into four distinct modules: the HAT, TAF, SPT, and DUB module. Bioinformatic analysis indicated that Fg contained the HAT, TAF, and SPT modules of the SAGA complex, although their amino acid sequence identities to corresponding homologs in *S. cerevisiae* were relatively low (Supplementary Table 7). In addition, the yeast Sus1p and Sgf11p orthologs were not identified in Fg, indicating that the components and amino acid sequences of the Fg SAGA are obviously different from those of *S. cerevisiae*, and which presumably leads to the different modes of PCN action against the budding yeast and filamentous fungi. SAGA mediates histone acetylation of gene promoters to enhance transcriptional

**Fig. 5** SAGA complex is essential for DON biosynthesis and virulence in Fg. **a** Pathogenicity of six gene deletion mutants of the SAGA complex on wheat heads. **b** Toxisome formation in the hyphae of ΔFgPCN, ΔFgADA3, and ΔFgGCN5 and wild-type strains under PCN treatment at 15 and 25 μg ml⁻¹. Tri1-GFP was used as the toxisomal marker for observation. The chemical fungicide carbendazim (abbreviated as Car) at the EC90 concentration, 1.4 μg ml⁻¹, was used as a control treatment; bar = 10 μm. **c** Expression of Tri1-GFP in tested mutants and the wild type treated with PCN or Carbendazim was verified by immunoblot assays using the anti-GFP antibody. In addition, the protein samples were also detected with the monoclonal anti-GAPDH antibody as a reference. **d** Deoxynivalenol (DON) production in the mutants and the wild type treated with PCN or carbendazim. DON in cell-free supernatants after 4 days of incubation in TBI medium were harvested, extracted, and quantified by LC-MS. Data presented are the mean ± s.d. (*n* = 3). **e** Toxisome formation in the hyphae of ΔFgPCN, ΔFgADA3, and ΔFgGCN5 mutants and wild-type labeled with Tri1-GFP inoculated on wheat leaf; bar = 40 μm. **f** Penetration structures of wild-type, ΔFgPCN, ΔFgADA3, and ΔFgGCN5 strains on dissected wheat glumes at 72 h post inoculation. Red arrows indicate the typical infection structures of the fungus

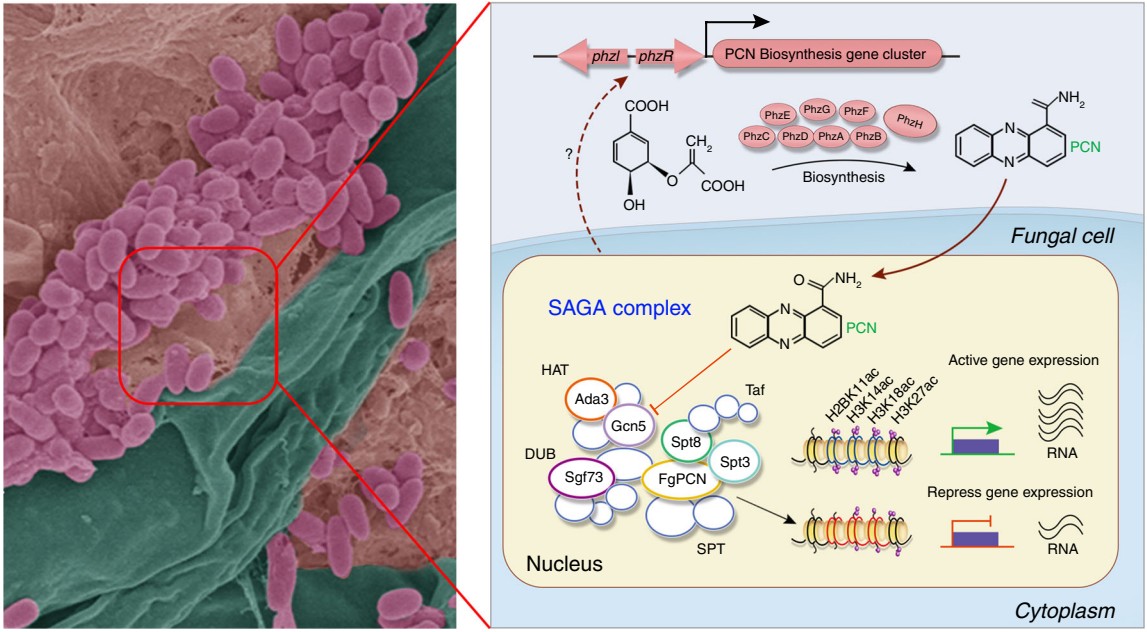

**Fig. 6** A model for the mode of action of PCN in ZJU60–Fg interactions. Biofilm formation of biocontrol agent *P. piscium* ZJU60 on the surface of fungal mycelia physically mediates the interaction between ZJU60 and Fg (left panel). Fungal mycelia, bacteria and matrix of bacterial biofilm are shown in green, purple, and brown, respectively, in the pseudocolor image. PCN biosynthesized by the *phzA-H* gene cluster is the major antifungal compound produced by ZJU60. In Fg, the SAGA complex regulates mycelium growth, deoxynivalenol (DON) biosynthesis, penetration and virulence mainly by modifying the histone acetylation of H2B11, H3K14, H3K18, and H3K27. During the interaction between ZJU60 and Fg, PCN diffuses into fungal cells and targets the SAGA complex by inhibiting the acetyltransferase activity of FgGcn5, which subsequently leads to reduced levels of histone acetylation, repression of gene expression, and suppression of fungal growth and pathogenicity. The number of purple circles on histone tails represents relative acetylated level of histones

activation by recruiting the pre-initiation complex and facilitates elongation by deubiquitinating monoubiquitinated histone H2B (H2B-Ub), and acetylating histones within the coding region to promote histone eviction. These multiple functions of SAGA contribute together to regulate the expression of target genes, and plays an essential role in developmental processes and environmental fitness in eukaryotes. In *S. cerevisiae*, the SAGA complex is required for the normal transcription of approximately 10% of genes (e.g., ribosome biogenesis, translation, amino acid metabolism), that are predominantly associated with growth and stress response[55]. Defects in hyphal development and increased sensitivity to cell wall stress were consistently observed in Δ*GCN5* mutants in dimorphic fungi, such as *C. albicans*[56] and *Ustilago maydis*[57]. In filamentous fungi, the SAGA complex is also critical for mycelial growth, asexual and sexual development, stress responses, and virulence. For example, disruption of the SAGA complex results in growth defects, a lack of asexual sporulation, and attenuated virulence in *A. nidulans*[58] and *Fusarium fujikuroi*[59]. Consistent with these previous reports from other fungi, we found that disruption of the SAGA complex led to significantly reduced mycelial growth (Fig. 4a and Supplementary Fig. 6), abolished conidiation in Fg. More importantly, we found that histone acetylation modified by the SAGA complex played an important role in secondary metabolite biosynthesis in Fg. Disruption of the SAGA complex via the deletion of genes encoding key components completely abolished DON biosynthesis (Fig. 5b–d). Consistently, the BCA ZJU60 exhibited a high efficacy in the management of FHB and DON production in both the growth chamber and field trials (Fig. 1c–e), unlike several commercial fungicides found to stimulate mycotoxin biosynthesis[16–18]. Thus, the BCA ZJU60 and its secondary metabolite PCN may possess a high potential for management of FHB and DON production, and the SAGA complex is a valuable target for the development of antifungal drugs.

The plant-associated microbiome serves as a protective barrier against invading pathogens both directly and indirectly[2,60,61]. However, little is known about microbe–microbe interaction dynamics in the microbial community. Here, we screened culturable beneficial bacteria from the wheat head microbiome and established a BFI system to understand the roles of commensal bacteria in the suppression of fungal disease. The present data suggest that increasing the population of the antagonist ZJU60 in the wheat head microbiome may suppress FHB disease. A comprehensive understanding of the mechanisms underlying the microbe–microbe interactions in the plant microbiome will provide new opportunities to control diseases and food safety.

## Methods

**Wheat head microbiome sequencing and analysis.** Healthy and natural Fg infected wheat heads (Cultivar: Yannong 19) were collected from Anhui province. Ten wheat heads were crushed and pooled as one sample, and three samples of healthy and infected wheat were collected and subjected to sequencing. DNA was extracted from 1 g of each sample using a Mag-Bind Soil DNA Kit (Omega, Norcross, GA) according to the manufacturer's instructions. The concentration and quality of each DNA sample were determined using a Nanodrop and agarose electrophoresis, respectively. Equimolar amounts of DNA were used as the template for PCR amplification. The V3–V4 hypervariable regions of the 16S rRNA gene were amplified from extracted DNA samples using two primers containing the universal sequences 338F (5′-ACTCCTACGGGAGGCAGCA-3′)[62] and 806R (5′-GGACTACHVGGGTWTCTAAT-3′)[63]. After amplification, the triplicate PCR products were pooled and purified using a PCR Cleanup Kit (Axygen Biosciences, Union City, CA, USA). PCR products were sequenced on a single lane of an Illumina MiSeq platform at Personal Biotechnology Co., Ltd. (Shanghai, China), and data were analyzed as described in supplementary methods.

**Isolation, identification, and antifungal activity of ZJU60.** Wheat head samples at 50 and 100% flowering from four different fields were collected for culturable bacterial isolation using the dilution plating method detailed in supplementary methods. At every stage, five samples were picked by a five-spot sampling method in a field, where ten healthy and infected wheat heads with a 50% disease index were included in each sample. Briefly, each sample was homogenized with a sterilized mortar and pestle. Macerated samples were serially diluted in sterile

0.85% NaCl solution, and the resulting suspensions were plated onto LB agar plates supplemented with the fungicide carbendazim to prevent fungal growth. Plates were incubated at 25, 30, and 37 °C for 1–3 days. A total of 12,854 (ZJU1-12854) culturable bacterial isolates were recovered from 80 independently pooled wheat head samples. Bacterial antagonistic activities towards Fg PH-1 were tested by dual-culture assay on WA[64], which support the vegetative growth of either bacteria or fungi. All isolates were tested in triplicate, and their inhibition zones were measured after 5 days of dual-culture at 25 °C. For ZJU60, its antagonistic activities towards seven other plant pathogenic fungi, *Alternaria alternata* Z7, *Botrytis cinerea* B05.10, *Fusarium oxysporum* Fo47, *Fusarium verticillioides* 7600, *Magnaporthe grisea* Guy11, *Penicillium italicum* PHI-1, and *Sclerotinia sclerotiorum* 1980 were also examined. All fungal strains were obtained from lab strain collection.

For molecular identification of the BCA ZJU60, the GC-FA and MALDI Biotyper was conducted as described in supplementary methods. The genome of ZJU60 was sequenced using the PacBio RSII sequencing platform. The phylogenetic tree depicting relationships between ZJU60 and the 22 completely sequenced representatives of *Pseudomonas* spp. (Fig. 1h) was constructed with ten housekeeping genes *16Sr DNA, aroE, dnaA, guaA, gyrB, mutL, ppsA, pyrC, recA,* and *rpoB* using the neighbor-joining method with Molecular Evolutionary Genetics Analysis version 4.0.

**Characterization of secondary metabolites in ZJU60.** Secondary metabolite gene clusters in ZJU60 were predicted using antiSMASH[65]. To identify antifungal compounds, ZJU60 was grown in KB broth for 4 days at 30 °C, and then cell-free supernatant was obtained by centrifugation. The supernatant was extracted three times with ethyl acetate (v/v = 1:1) and evaporated. Resulting crude extracts were subjected to silica gel column chromatography (CC) (200–300 meshes). Extracts were eluted with CHCl$_3$ followed by CHCl$_3$:MeOH at ratios ranging from 99:1 to 0:100. The antagonistic activity of each fraction was tested against Fg mycelial growth. Fractions eluted using CHCl$_3$:MeOH (85:15) displaying a high antifungal activity was further identified by a Bruker Avance DRX-500 MHz spectrometer with TMS as an internal standard[64].

Disruption of the PCN biosynthesis cluster in ZJU60 was performed using the suicide plasmid pEX18Tc and a two-step homologous recombination method[66]. Briefly, the homologous PCR products were ligated into digested pEX18Tc using the In-Fusion® HD Cloning Kit (Cat. No. 639649, TaKaRa Biotechnology Co., Ltd.). The recombinant plasmids were then transformed into *E. coli* WM3064 (2,6-DAP auxotroph) before being conjugated to ZJU60[67]. The double-crossover recombinants were screened on LB plates containing kanamycin and then verified by PCR and sequencing using specific primers.

**Construction of fungal deletion and complementation strains.** Gene deletion vector construction and transformation of Fg or *B. cinerea* were was generated by the double-joint (DJ) PCR method[31]. The primers used to amplify the flanking sequences for each gene are listed in Supplementary data 5. Each open reading frame (ORF) was replaced with a hygromycin resistance cassette (hyg) and subsequent deletion mutants were identified by PCR with specific primers (Supplementary data 5). For complementation, the respective ORFs were fused to a geneticin selection marker, and introduced into the corresponding deletion mutants.

The Tri-GFP fusion cassette in pYF11 was constructed previously[17]. Recombinant plasmid was then transformed into PH-1 or ΔFgPCN, ΔFgADA3, and ΔFgGCN5 mutants to generate fluorescent labeled strains to visualize toxisome formation under DON inducing conditions and on wheat leaves.

**Evaluation of biocontrol efficiency.** The biocontrol efficacy of ZJU60 against FHB in a growth chamber was determined using the wheat cultivar Jimai 22 (susceptible to FHB) and the protocols described previously[64]. Briefly, mid-anthesis stage wheat heads grown in a growth chamber were sprayed with a cell suspension of biocontrol strain ZJU60 (10[8] CFU per ml) using a hand-held atomizer. The fungicide phenamacril (2 mg ml$^{-1}$) and sterile water alone served as positive and negative controls, respectively. After the application of prevention treatments for 6 h, each treated wheat head was sprayed with a 3 ml of a conidial suspension of Fg strain PH-1 at a concentration of 10[4] conidia per ml with 0.05% Tween 20. After 7 days of incubation in the growth chamber, the infected wheat heads were imaged. A total of 25 wheat heads were used for each treatment, and the experiment was repeated twice.

Field experiments were conducted at Huai'an, Jiangsu Province and Hangzhou, Zhejiang Province in China in 2015 and 2016 with wheat cultivars Yangmai 18 and Jimai 22, respectively. The field trials were conducted using a randomized plot design with three replicates per each treatment. Each plot was 100 m$^2$ in size. Each plot was sprayed twice at 5 and 15% anthesis with 2.5 l of ZJU60 cell suspensions at the concentration of 10[8] CFU per ml amended with 0.05% Tween 20. Fungicide phenamacril and sterile water alone served as positive and negative controls, respectively. Twenty-five days after inoculation, FHB disease severity in each plot was examined. Disease index and efficiency were assessed as described[64] (see details in the supplementary methods).

**Western blot assay.** Protein extraction and western blot analysis for detecting histone acetylation profiles or Tri1-GFP expression of the wild-type strain and ΔFgGCN5 with or without PCN treatment were performed as described previously[31], and as detailed in the supplementary methods. The experiment was conducted three times independently. All blots were imaged by the ImageQuant LAS 4000 mini (GE Healthcare) (Supplementary Fig. 12).

**Pathogenicity assays and DON biosynthesis assays.** Given that mutants of the SAGA complex were defective for conidiation, 0.1 mg of fresh mycelia from each strain was used to inoculate the middle spikelet of flowering wheat heads. Twenty individual wheat heads were inoculated for each tested strain. Images were taken at 15 days after inoculation.

To observe toxisome formation patterns following treatment with PH-1, derived mutants and PCN, all strains were labeled with FgTri1-GFP and cultured in TBI liquid medium[68] for 48 h before observation with a confocal microscope. To quantify DON production in the wild type, mutants and PCN treatments grown in TBI liquid medium, DON was extracted from each strain after incubation for 4 days. The cell-free supernatant was filtered and passed through a SampliQ Amino (NH2) solid phase extraction columns (Agilent Technologies), and 4 ml of the purified extract were evaporated to dryness under a nitrogen stream. The residue was dissolved in 1 ml methanol:water (40:60, v/v), followed by centrifugation at 10,000 rpm and subsequently analyzed by LC-MS/MS[17].

**Biofilm formation assay.** To test the ability of ZJU60 to form biofilm, ZJU60 cells were first grown in 3 ml of KB broth to OD600 = 1.0. Then, culture (3 μl) was added to 3 ml of WA or YEPD in 12-well polyvinyl plates. The plates were incubated statically at 30 °C for 72 h to evaluate pellicle formation. Submerged biofilms were quantified by CV staining as follows[25]. After 72 h of growth at 30 °C, the medium in each well was gently removed with a pipette and each well was washed twice with 1 ml of phosphate buffered saline (PBS) to remove unattached bacteria. The plates were stained with a 0.1% CV solution (Sigma) for 30 min and then washed five times with 2 ml of PBS. Plates were air dried before imaging. The CV was then resuspended with 50% acetic acid, and diluted 10 fold, and the absorbance was measured at 570 nm. The PsI biofilm matrix produced by ZJU60 and ΔphzA-H was quantified by immunoblotting with anti-Psl serum[27] (details in supplementary methods).

Biofilm formation by ZJU60 or ΔphzA-H on the mycelia of Fg PH-1 was visualized by SEM. PH-1 was first grown in 150 ml of WA liquid medium for 18 h at 25 °C, and then 1.5 ml of ZJU60 or *ΔphzA-H* cells (OD600 = 1.0) was added to the flask. The co-cultured medium was incubated statically at 25 °C for 5 days. After incubation, the mycelia were harvested and prepared for SEM observation.

**Bacterial colonization on fungal mycelium.** Six co-culture duplications were carried out for each treatment (ZJU60 vs Fg or ΔphzA-H vs Fg) at each time, and the experiments were repeated three times. To quantify the total bacterial cells after 5 days of co-culture, three flasks in each treatment were rapidly vortexed, then the cultures were filtered through layers of gauze to exclude the mycelia. The mycelia on the gauze were washed three times with PBS. All culture filtrates were collected and serially diluted for counting. To quantify the mycelium-associated bacterial cells, the co-cultures in the remaining three flasks per treatment were not vortexed and were filtered through gauzes to collect the mycelia. The mycelia were gently washed with PBS to exclude non-associated bacterial cells. Then, the harvested mycelia were rapidly vortexed in PBS, and the supernatants were subsequently used for assays.

**In vitro acetylation assay.** An in vitro acetylation assay was performed as previously described[69] with minor modifications. Briefly, purified GST-H3 protein was incubated with FgGCN5-His protein in HAT buffer (50 mM Tris–HCl [pH 8.0], 50 mM KCl, 0.1 mM EDTA, 1 mM DTT, 1 mM protease inhibitor, 5% glycerol, and 5 mM acetyl-CoA) in the presence or absence of 30 μg ml$^{-1}$ PCN for 45 min at 37 °C. The reactions were resolved by SDS-PAGE and analyzed by western blotting with H3ac (pan-acetyl) (#39139, 1:50000) and H3K18ac (#39756, 1:7500).

**Determination of PCN–FgGcn5 binding by SPR.** The affinity of PCN for FgGcn5 was measured by direct SPR assays using the mode of multiple cycle kinetics in the Biacore T200 system (GE Healthcare). The purified FgGcn5 was diluted in 10 mM sodium acetate buffer (pH 4.0) to a final concentration at 50 μg ml$^{-1}$, and covalently immobilized onto a CM7 sensor chip (cat. 28-9538-28, GE Healthcare) via amine coupling chemistry. A series of PCN solutions at different concentrations (0, 1.25, 2.5, 5, 10, 20 nM) were created using running buffer (10 mM PBS, pH 7.4, 150 mM NaCl) with an equal volume 5% DMSO. The solutions were injected at a flow rate of 30 μl min$^{-1}$ for 120 s, followed by dissociation for 180 s. After each binding reaction, the immobilized surface was regenerated with the injection of 10 mM glycine–HCl (pH 2.0) at the flow rate of 10 μl min$^{-1}$ for 60 s. In addition, solvent correction was performed by injecting eight solutions of running buffer containing 5% DMSO. To correct for the DMSO effect, solvent correction was applied before proceeding with the evaluation. The corrected response (reference subtracted and solvent corrected) from the active surface vs cycle number was obtained. To evaluate the results, the response of the blank control (zero

concentration) was also deducted from that of each sample containing PCN, and the blank-deducted sensorgrams were subjected to kinetic fitting using Biacore Evaluation Software 3.0.

**Data analysis**. All statistical analyses were performed using SAS software. Data are presented as mean ± standard deviation (s.d.). Differences between two groups were analyzed by Student's $t$ test. Multiple comparisons were analyzed by one-way analysis of variance (ANOVA) followed by least significant difference (LSD) multiple-range test. The normality of data was tested using the Kolmogorov–Smirnov test ($P < 0.05$), and the equality of error variances was tested by Levene's test ($P < 0.05$). In the case of nonnormality and/or unequal variances, data were transformed before ANOVAs.

**Data availability**. The genome sequence of ZJU60, the wheat microbiome data, and the chemo-genomic profiling data have been deposited in the NCBI BioProject databases with accession codes PRJNA436763, PRJNA473402, and PRJNA395460, respectively. Other relevant data supporting the findings of the study are available in this published article and its Supplementary Information files.

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

## Acknowledgements

We thank Professor Shengyang He (Michigan State University) for manuscript editing and helpful suggestions. We thank Professor Tingjun Hou and Weitao Fu (Zhejiang University) for the molecular docking analysis and Professor Lvyan Ma (Chinese Academy of Sciences) for the anti-PsI serum. This research was supported by the Natural Science Foundation of Zhejiang Province for Distinguished Young Scholar (LR17C140001), the National Natural Science Foundation of China (31525020 and 31672064), the China Agriculture Research System (CARS-3-1-15), the Young Elite Scientist Sponsorship Program (2017QNRC001), the International Science & Technology Cooperation Program of China (2016YFE0112900), the National Key Research and Development Program of China (2016YFD0300706 and 2017YFD0201104), and the Dabeinong Funds for Discipline Development and Talent Training in Zhejiang University.

## Author contributions

Y.C., Y.r.C., and Z.M. designed the experiments; Y.C., J.W., N.Y., Z.W., and X.S. performed the experiments; Y.C. and J.W. analyzed the data; Y. C., Y.r.C., and Z.M. wrote the manuscript.

## Additional information

**Competing interests:** The authors declare no competing interests.

