## [Peer Review File · Nature Communications]

Reviewers' comments:

Reviewer #1 (Remarks to the Author):

In the paper entitled "epigenetic regulation mediated by a member of the wheat head microbiome reduces virulence and growth of a major wheat fungal pathogen", the authors reveal a new form of fungal killing by a bacterial compound. In the manuscript the characterization of the bacterial-fungal interaction is well documented, however the epigenetic regulation needs to be reinforced. Additionally, the authors do not fully address the fungal mutants growth defects, nor do they provide strong evidence that the FgPCN locus is directly involved with sensitivity – they at best demonstrate that this locus controls additional components essential for fungal growth and/or response to PCN. Specific comments can be found below:

- The results sections part 1, 2 and 3 are very strong, interesting and represent a large body of carefully carried out experiments.
- In the results part 4, the reasoning behind the mutant screen is unclear. Usually one looks for a rescue phenotype rather than a hypersensitivity phenotype. With a hypersensitivity phenotype, it is unclear how the authors can claim that the mutant is in the same pathway as the effect of PCN on growth.
- In general the issue of growth defects in mutants has not been addressed carefully enough. The authors state in lines 300: "To verify that the PCN sensitive phenotype of Δ FgPCN was not caused by slow mycelial growth of the mutant, we extended the incubation time to 6 days until the mycelia of Δ FgPCN without PCN treatment reached the edge of plates." This argues that the mutant has a growth defect as the wildtype PH-1 grows to cover the plate within 3 days in the DMSO control group, whereas the mutants do not.
- Along with the previous comment, the statement on line 304 is an overstatement: "Meanwhile, the complementation strain (designated Δ FgPCN-C) fully rescued for the PCN susceptibility (Fig. S4b). Therefore, these results strongly support the idea that the FgPCN gene is associated with the PCN susceptibility in *F. graminearum*." Indeed, as the mutant alone is defective for growth, the hypersensitivity to PCN is probably a reflection of the growth defect in untreated conditions.
- Similarly, I do not agree with the statement on line 327 "these data indicated that the SAGA complex has a role in PCN susceptibility" since the mutant alone is defective for growth, the hypersensitivity to PCN is probably a reflection of the growth defect in untreated conditions, and does not reflect the link between PCN susceptibility and the SAGA complex.
- The authors state in lines 358-365: "We next tested whether the SAGA complex was also associated with the susceptibility of *F. graminearum* to other antifungal drugs. Three widely used fungicides, fludioxonil, phenamacril and triadimefon, were selected. These compounds target the MAPK signal transduction, motor protein myosinI, and sterol biosynthesis, respectively. As shown in Fig. S5, deletion mutants of key SAGA components did not alter the sensitivity towards these antifungal drugs. These results demonstrated that the SAGA complex may be associated with the PCN susceptibility specifically in *F. graminearum*." Here again, there is a misinterpretation of the data, as the mutants compared to wildtype, and to the mutants grown on control media there are apparent differences in the growth at 3 days. Additionally there is no image representing PH-1 at 5 days. These further suggest that the mutants impact the fitness of the organism in general, especially without complementation of the genes for comparison.
- The western blot in figure 4 (which is mislabeled e. compared to the legend f.) clearly shows that GCN5 is important for histone acetylation, and suggests that PCN has an effect on histone acetylation. However this figure does not show that PCN affects GCN5 as other HATs or HDACs could be targeted by PCN.
- The activity assay in figure 4 (mislabeled f. rather than g.) clearly shows that PCN inhibits GCN5, but this does not show that it is the effect of PCN on GCN5 that blocks histone acetylation. Therefore the

statement on p 13 “collectively, our evidences indicate that PCN specifically targeted the histone acetyltransferase” is an over statement.

- On line 362 the authors state that fig. S5 shows that deletion mutants of SAGA did not alter the sensitivity towards the antifungal drugs. However the mutants do look attenuated. Is there a mistake?

- In the results section part 5 it is unclear how any conclusion can be made on the role of the SAGA complex on pathogenicity if the SAGA complex is important for growth.

- In general the paper needs to be edited both for content, mainly figure legends, and for English grammar which is lacking throughout.

Reviewer #2 (Remarks to the Author):

This is a very interesting, comprehensive and novel study. The initial approach used to find a novel biocontrol strain is very traditional. But then the researchers undertake a series of both 'brute force' and elegant experiments to define the mechanism conferring the inhibitory effects of the *Pseudomonas* strain ZJU60 against *F. graminearum*. Overall, the manuscript is well presented, well written, laid out and the figures and tables are high quality.

This said, there are still several major and more minor concerns, which the authors need to address.

Major concerns

The formation of biofilms and pellicles

Abstract Line 20-21 states 'ZJU60 forms biofilms on the hyphae of *F. graminearum*',. This reviewer struggled to find evidence in the manuscript to support this strong statement. The evidence in the results section is as follows Lines 222 – 225 '...observed with scan electron microscopy. After 5 days of co-culture, a large number of ZJU60 cells formed extensive biofilm-like structure on the surface of fungal mycelium and caused the shrivelled mycelium (Fig. 3d)'. There must be biochemical markers for biofilm formation. The authors cannot rely just on SEM observations on deformed mycelium and the presence of bacteria. Or the authors need to drop entirely the concept of a biofilm.

Similarly, the detection of pellicle and biofilm formation in another experiment
Line 243 – the word 'pellicles' needs to be defined and a suitable reference provided

Line 243 – 245 states 'JU60 cells formed thin floating pellicles at the air-liquid interface (Fig. 3h, right-hand panels). Meanwhile, cells also formed submerged biofilms in the WA and YEPD media, especially the WA medium (Fig. 3h, top panels). There is no biochemical evidence for either floating pellicles or submerged biofilm formation. These are key points that need to be addressed via additional experimentation.

And in another experiment

Line 255-256 In addition, ZJU60 was also form biofilm on the surface of wheat tissues. Again no biochemical evidence for biofilm formation.

Lines 757-758 Submerged biofilms were quantified by crystal violet staining as follows. Is there a reference for this method of quantification of submerged biofilms? If not how do the authors know that only biofilms are detected by crystal violet staining. Crystal violet is a very general stain used for multiple purposes.

Finally, in the discussion, there is no mention of biofilm formation which tends to suggest that the authors do not themselves think this is a very strong part of their study.

The microbiome

Lines 125-126 – “It was implied that wheat head might recruit antagonistic bacteria in microbiome for defending infection by the fungus.” – This is a very strong statement for the minimal evidences shown to this point. Therefore, this point should be stated as a hypothesis, not implied based on the data shown so far.

PCN induction and the eventual model

Lines 217 -221. ‘We extracted and quantified the PCN production in the ZJU60 culture alone and co-culture with *F. graminearum* using LC-MS. As shown in Fig. 3c, the relative PCN production was increased about 50% in co-culture than the ZJU60 culture alone. These data suggested that physical interaction between ZJU60 and *F. graminearum* stimulated the production of the antifungal compound PCN’. This is an exceptionally interesting result. But it is also a very counterintuitive one. This result suggests that the presence of wild-type *F. graminearum* hyphae leads to the strong induction of a compound by ZJU60 which will have a strong detrimental effect on fungal hyphae health. There would be very strong evolutionary pressure within *F. graminearum* to mutate and not to be an inducer of PCN. This point needs to be considered in the manuscript discussion. This PCN induction data also strongly impacts on the working model given in Figure 6 and has not yet been considered.

Fg mutant analysis

An impressive number of single gene deletion Fg mutants were tested. Some have previously been published, others have not. In lines 296-297 the following are stated ATP-binding cassettes (60 genes), transcription factors (105 genes), and epigenetic modification related proteins (72 genes)’. But no other information is given on exactly the Fg locus ID evaluated. For completeness an additional supplementary file needs to be included giving the IDs of this additional set of 237 genes. This is valuable negative information that can be used in bioinformatics/ network analyses and also indicates to the global fungal community that these mutant strains are available for both within and cross species comparisons.

DON mycotoxin induction

Lines 401 – 404 states ‘In addition, DON is synthesized in the specific cellular compartmentalization toxosome in *F. graminearum*, which is mainly formed by trichothecene biosynthesis enzymes encoding by TRI genes 35, 36. The experiment under evaluation is a wheat head time course involves predominantly extensive in planta colonisation. The two papers cited reveal that the toxosome structure is occurring at the immediate initial post penetration phase or in vitro. There is no evidence to link toxosome formation to any of the later stages of the infection process in the peer reviewed literature. This text in the manuscript will need to be substantially revised to provide the necessary explanation of the DON results provided.

Lines 405-406 ‘we examined toxosome formation by measurement of Tri1-GFP, a key DON biosynthetic enzyme’ Exploring the induction of Tri 1, which does not reside in the main TRI biosynthetic cluster on Chromosome 2 is a strange choice to explore DON induction. The authors need to give the rationale and cite a reference(s) for why this gene choice is suitable / optimal to explore

DON induction.

Lines 418 – 421 states PCN at either EC50 or EC90 significantly suppressed Tri1-GFP expression, disrupted the toxosome formation, and reduced the DON production in the wild type strain, while carbendazim enhanced the DON biosynthesis (Fig. 5b, c, d). Is the carbendazim result confirming what has already been reported in the literature, or is this the first evidence that another fungicide triggers DON induction? Please clarify for the reader.

Comparing mutant phenotypes

Lines 515-517 Due to the fact that DON is responsible for virulence of *F. graminearum*, all SAGA complex mutants had a complete loss of pathogenicity in planta (Fig. 5a). But this argument is somewhat flawed because neither the Tri1 or Tri5 mutants exhibit such a severe reduction in pathogenicity towards wheat heads at the SAGA mutants. This comparative loss of disease causing ability between mutants needs to be further discussed.

Figures – major points of concern

Figure 1a – The image of bacteria colonies with green crystals is in very poor quality

Figure 1 – d. The data needs to be presented as absolute values and not as shown comparative data. The reader has no idea of the actual disease pressure in these two field trial years. This will be very important for comparative analyses in the future across treatment types and years.

Figure 2 Panel c. There is an excessive amount of aerial mycelium extending from two or the three wheat ears. This is highly unusual when the heads remain attached to the plants and are photographed immediately after detachments. Either the plants were re-placed in a high humidity box for 24 hr prior to the image capture or the heads were detached and placed in high humidity for a period of time before image capture. Please clarify exactly how this image was obtained. This normal types of fusarium infected wheat heads are shown in Figure 5 panel a.

Figure 3 Panel a. Very poor quality, needs to be replaced with far better images.

Figure 4 – Figures 'e' and 'f' does not match what is in the text and the indication for figure panel 'g' is missing.

Lines 183-187 – “In contrast, the growth inhibitory activities of the mutants that had disrupted biosynthetic operons for hydrogen cyanide, pyoverdine and achromobactin, respectively, were not significantly reduced, when compared to that of the wild type ZJU60 (Fig. 2a)” – Fig. 2a does not show any data related to this statement. It would be good to have in the supplementary material the inhibitory activity of other mutants.

Discussion

Phenazines are well known determinants of biological control of soil-borne plant pathogens by various strains of fluorescent *Pseudomonas* spp (Doi 10.1046/j.1469-8137.2003.00686) and have also been tested in the past against *Fusarium oxysporum* (doi: 10.1016/j.micres.2010.06.001). Therefore, phenazines should be discussed in this wider context.

Other changes required

Additional missing references and / or poor explanations and / or incorrect information

Lines 93-95 – “Unfortunately, fungicide-resistant *F. graminearum* strains have been detected in the field after long-term intensive application of the fungicides.” - missing reference.

Lines 95-96 states ‘Moreover, several fungicides were able to stimulate the mycotoxin production’. A more balanced sentence with additional references is required here because some fungicides do not appear to directly induce DON mycotoxin accumulation whereas others do.

Missing technical information

Line 136 – Was the strain ZJU60 recovered from healthy or infected wheat heads?

Lines 143-144 – “Similar to the fungicide phenamacril, the treatment of ZJU60 by foliar spray...” – In this sentence or in the introduction, it would be worthy to mention that phenamacril is a fungicide widely used to control FHB in China

Lines 543 – 544 Healthy and natural *F. graminearum* infected wheat heads (Cultivar: Yannong 19) were collected from Anhui province. Is the cultivar Yannong semi-resistant or fully susceptible to FHB?

Line 560-561 – which stages? How many samples were from healthy and infected?

Lines 574 -576 For ZJU60, its antagonistic activities towards 7 other plant pathogenic fungi including *Alternaria alternata*, *Botrytis cinerea*, *Fusarium oxysporum*, *Fusarium verticillioides*, *Magnaporthe grisea*, *Penicillium italicum*

576 and *Sclerotinia sclerotiorum* were also examined. All these strains need references and / or full details if this is their 1st publication. Please provide as a supplementary table.

Line 605 antiSMASH – missing reference

Ln 659-661: “The primers used to amplify the flanking sequences for each gene are listed in Table S1.” This table is currently missing. Instead Table S1 is “Average relative abundances of the dominant genera in the wheat head microbiome “

Line 674 Recombination plasmids – what selectable marker was used for the *Fg* transformations, this is not obvious from the way this paragraph is constructed.

Line 719 tested strains were grown in TBI medium. What is the full composition of TBI media?

Line 729-732 – The reference Liu et al., 2015 describe the pathogenicity assay using *Fg* spores. If they used mycelia, how mycelia concentration was measured and how much was used?

Minor points

Lines 48 and 52. The acronym SGAG is incorrect and should be changed to SAGA

Line 65. ‘compositions’ a better word choice is required

Line 114 - In the rhizosphere, increasing evidences have plants are able to recruit.... Sentence structure does not seem to be right, rewrite

Lines 124 – if more than one species of *Pseudomonas*, it should be *Pseudomonas* spp.

Lines 129-130 . 'Therefore, we are toward to use a simple cultivable bacterial-F. graminearum interaction system...'. Poor English , sentence needs to be rewritten.

Line 393 Due to *F. graminearum* is a plant pathogen,... Poor English , sentence needs to be rewritten.

Line 574 – 'alternat' should be 'alternata'

Line 575 – 'grisea' should be in italic

Reviewer #3 (Remarks to the Author):

Comment to the manuscript entitled "Epigenetic regulation mediated by a member of the wheat head microbiome reduces virulence and growth of a major wheat fungal pathogen"

The manuscript describes a novel mechanism for *Fusarium* disease control by *Pseudomonas*. The manuscript provides a comprehensive dataset, and the rationale behind the experiments is well described. However, in multiple occasions, the description and the presentation of the data should be improved substantially to meet the quality of the journal. See further the comments below.

The introduction is poor. The English writing needs improvement. The authors refer to several examples of bacterial fungal interactions, but many more have been described. The authors state several times that the molecular mechanism behind bacterial-fungal interactions are unknown, but they have been described for several systems. Either the authors are unaware, or perhaps trying to say something else that is not coming across. Since the authors present a new type of mechanism of how a potential biocontrol bacterium can control a pathogenic fungus, the authors should include the different mechanisms that have been described so far for bacteria-fungal interactions to control fungal plant pathogens (for example antimicrobial compounds and volatiles, mycophagy, induction of plant resistance, etc) so they can better emphasise the novelty of their finding. In the results section, several statements are made that are incorrect, see detailed comments below.

An impressive array of analyses have been performed, including microbial community analysis of wheat head, bacterial isolations, *Fusarium* growth inhibiting assays, fatty acid analysis, biocontrol assays in climate chamber and field, phylogeny of concatenated housekeeping genes, genome sequencing, mutagenesis of *Pseudomonas*, LC-MS, SEM and TEM, colonization and biofilm formation of *Pseudomonas*, mutagenesis of multiple pathways in *Fusarium* (ROS-associated, SAGA, acetyltransferase), mutagenesis of *Botrytis*, and *Magnaporthe*, mutant library screenings of *Fusarium* and *Saccharomyces*, immunoblotting, histone acetyltransferase inhibition, toxosome formation. Even though the paper describes this large amount of data, the presentation of the data is sometimes unclear, and the result sections should be substantially revised. Figures should be carefully checked for spelling errors. Also figure legend and/or methods section should be transparent and detailed. Often statistical analyses, what error bars indicate etc are not described.

Detailed comments

In various sections the English spelling is incorrect or unclear, a few are included below.

L114: incomplete sentence

L117: 16S sequences cannot give information about the protective status. Rephrase.

L119: mention how many OTUs were obtained and how many families/genera.

L125: I don't think it is just to say that this is 'recruitment'. It could also be a response of the community because the wheat head environment is changing because of the infection and might leak other nutrients. I suggest rephrasing

L129: unclear sentence

L133: infected or uninfected? Did the author focus on a specific family of bacteria based on the microbiome analysis? Did they use specific media?

L137: on top of its colony.

L143: 'material and methods' is labelled as 'method'

L153: Please indicate whether these are high scores for the identification. Not all readers are aware if these values are good or not.

L175-176: how are these genes distinctly different? This cannot be seen in Table S5. It would be helpful to add identity percentages with known HCN, PCN, PCA and achromobactin genes.

L187: fig 2a is not referring to the data described in the sentence.

L210: how do authors know that the green substance is PCN?

L221: what about PCA production?

L225: it is not proven that the biofilm formation of the bacteria is causing the shrivelled mycelium

L229: Fig S2a is not showing the Δ phzA-H vs *F. graminearum* co-culture. Spelling *F. graminearum* incorrect.

L235: unclear how total culturable cells and specific bacteria have been obtained. Typing error y-axis 3g.

L251: Fig 3i showing SEM not present.

L263: what is meant with "chemical genomic analysis"?

L270: what is meant with "interacting mutants"?

L309: It can be debated whether Spt7 is really homologues with only 20% identity as described in L268. Was the Spt7 mutant of *S. cerevisiae* screened in the mutants described in L263-270? If so, could the authors explain why this mutant did not show a change in susceptibility to PCN?

L367: when comparing the protein sequences of Spt7 and FgPCN, can the authors find a reason for the clear separation between fungi and yeast homologs? Are there any domains different?

L393: unclear English

L401: unclear what is meant with 'a spike'

L452: Is *Listeria* the only pathogenic bacteria described for this phenomenon? Describe a few.

L486: be consistent in naming of H3K27 etc, it is different here compared to L334 and in Fig 4.

L516: Since it is apparently easy to construct mutants in *Fusarium*: has a TRI-mutant lost its virulence?

L558: please describe the methods used for analysing the sequencing data. The reference provided is not describing 16S amplicon data analysis.

Fig 1f: what each ring in the genome is representing should be described in figure legend.

Fig 2: order of panels is not logic. (2d above b/c). In 2g: include pictures of the control (0 ug/ml).

Table S4: include the accession numbers of the other *P. chlororaphis* isolates.

Fig S4b: present data in graph by measuring hyphal diameter.

Response to referees

Reviewers' comments:

Reviewer #1 (Remarks to the Author):

In the paper entitled “epigenetic regulation mediated by a member of the wheat head microbiome reduces virulence and growth of a major wheat fungal pathogen”, the authors reveal a new form of fungal killing by a bacterial compound. In the manuscript the characterization of the bacterial-fungal interaction is well documented, however the epigenetic regulation needs to be reinforced. Additionally, the authors do not fully address the fungal mutants growth defects, nor do they provide strong evidence that the FgPCN locus is directly involved with sensitivity – they at best demonstrate that this locus controls additional components essential for fungal growth and/or response to PCN. Specific comments can be found below:

- The results sections part 1, 2 and 3 are very strong, interesting and represent a large body of carefully carried out experiments.

Re: Thanks.

- In the results part 4, the reasoning behind the mutant screen is unclear. Usually one looks for a rescue phenotype rather than a hypersensitivity phenotype. With a hypersensitivity phenotype, it is unclear how the authors can claim that the mutant is in the same pathway as the effect of PCN on growth.

Re: We agree with the reviewer that researchers often look for resistant phenotype rather than a hypersensitive phenotype to identify the potential drug target. However, the deletion mutants of drug target genes sometimes also result in the hypersensitivity phenotype. For example, the azole fungicides directly inhibit C14 α -demethylase encoding by *CYP51* gene(s), and block the ergosterol biosynthesis to suppress the fungal growth. In two previous studies, we found that the deletion mutant of the azole drug target Δ CYP51A resulted in a hypersensitive phenotype towards azole fungicides in both *Fusarium graminearum* [Fungal Genetics and Biology 48 (2011) 113–123] and *Magnaporthe oryzae* [Fungal Genetics and Biology 48 (2011) 144–153]. Therefore, it is logical for us to propose that the SAGA complex might be involved in the same pathway as the effect of PCN on growth, according the hypersensitive phenotype of SAGA mutants towards PCN. Additionally, the PCN treatment reduced the level of histone acetylation, which was catalyzed by the SAGA complex in *F. graminearum* (Figure 4e). The new data provided in this revised manuscript further showed that PCN is able to inhibit the

acetyltransferase activity of FgGcn5 in the *in vitro* assay (new data in Figure 4f) and strongly bind to FgGcn5 with $K_D=71.37$ nM (new data in Figure 4g). Furthermore, the binding site (or pocket) of PCN in the HAT of FgGcn5 was predicted by the multiple ligand simultaneous docking (MLSD) program, and the results showed that binding of PCN to FgGcn5 creates a relatively stable PCN-FgGcn5 complex (new data in Fig. S8, Video S1-3).

- In general the issue of growth defects in mutants has not been addressed carefully enough. The authors state in lines 300: "To verify that the PCN sensitive phenotype of Δ FgPCN was not caused by slow mycelial growth of the mutant, we extended the incubation time to 6 days until the mycelia of Δ FgPCN without PCN treatment reached the edge of plates." This argues that the mutant has a growth defect as the wildtype PH-1 grows to cover the plate within 3 days in the DMSO control group, whereas the mutants do not.

- Along with the previous comment, the statement on line 304 is an overstatement: "Meanwhile, the complementation strain (designated Δ FgPCN-C) fully rescued for the PCN susceptibility (Fig. S4b). Therefore, these results strongly support the idea that the FgPCN gene is associated with the PCN susceptibility in *F. graminearum*." Indeed, as the mutant alone is defective for growth, the hypersensitivity to PCN is probably a reflection of the growth defect in untreated conditions.

- Similarly, I do not agree with the statement on line 327 "these data indicated that the SAGA complex has a role in PCN susceptibility" since the mutant alone is defective for growth, the hypersensitivity to PCN is probably a reflection of the growth defect in untreated conditions, and does not reflect the link between PCN susceptibility and the SAGA complex.

Re: The rate of radial growth of Δ FgPCN and other mutants of SAGA components was reduced appropriately 40-63% in comparison with that of the wild type on the PDA medium without PCN. To exclude the possibility that the hypersensitivity phenotype towards PCN may be caused by growth defect, we extended the incubation time until the mycelia of mutants without PCN treatment reached the edge of plates. As results shown in Figure S6 and the new data in Figure S9, Δ FgPCN was still not able to grow on the PDA medium with PCN, and the growth inhibition rate was near 100% (Figures 4b, S9b). These results suggested that the PCN hypersensitive phenotype of Δ FgPCN could not be primarily caused by slow mycelial growth of the mutant.

In addition, lots of mutants in our deletion mutant library show a growth defect, but they have almost the same sensitivity towards PCN as the wild type. For example, deletion mutants of Δ FgCHS7 (locus FGSG_12039) and Δ FgAP2

(locus FGSG_02105) showed similar or more severe growth defect, compared that of $\Delta FgPCN$ on PDA medium. However, these two mutants did not increase the sensitivity towards PCN as compared to $\Delta FgPCN$. In summary, our evidence suggested that the growth defect of mutants is not associated with the sensitivity towards PCN in *F. graminearum*. We do agree however that the growth defect could marginally contribute to the hypersensitivity phenotype.

- The authors state in lines 358-365: “We next tested whether the SAGA complex was also associated with the susceptibility of *F. graminearum* to other antifungal drugs. Three widely used fungicides, fludioxonil, phenamacril and triadimefon, were selected. These compounds target the MAPK signal transduction, motor protein myosinI, and sterol biosynthesis, respectively. As shown in Fig. S5, deletion mutants of key SAGA components did not alter the sensitivity towards these antifungal drugs. These results demonstrated that the SAGA complex may be associated with the PCN susceptibility specifically in *F. graminearum*.” Here again, there is a misinterpretation of the data, as the mutants compared to wildtype, and to the mutants grown on control media there are apparent differences in the growth at 3 days. Additionally there is no image representing PH-1 at 5 days. These further suggest that the mutants impact the fitness of the organism in general, especially without complementation of the genes for comparison.

Re: Thanks for pointing out this. We agree with the reviewer and we believe that it is very important to exclude that the hypersensitivity phenotype of SAGA mutants towards PCN is a general feature of fitness defect. Therefore, we tested the sensitivity of SAGA mutants towards other fungicides. In the revised manuscript, as suggested by the reviewer, the complemented strains were included as controls. The larger petri dishes (diameter =15 cm) were used for the wild type PH-1 and complemented strains, while the 6-cm petri dishes were used for mutants. The images taken after 6 days of incubation were shown in Figure S9a in this new submission.

The inhibition of mycelial growth rates (%) by each fungicide towards fungal strains was statistically analyzed. As shown in Figure S9b, deletion mutants did not significantly increase their sensitivity to other tested fungicides including fludioxonil, phenamacril and triadimefon. Therefore, our data suggested that the SAGA complex may be specifically associated with the sensitivity towards PCN, but not other tested fungicides in *F. graminearum*.

-

- The western blot in figure 4 (which is mislabeled e. compared to the legend f.) clearly shows that GCN5 is important for histone acetylation, and suggests that PCN has an effect on histone acetylation. However this figure does not show that PCN affects GCN5 as other HATs or HDACs could be targeted by PCN.

Re: We fixed the label in the figure legend. In our survey, a total of 31 predicted acetyltransferase genes were found in this fungus, and each one of them was disrupted by the homologous recombination. Among the resulting mutants, only Δ FgGCN5 showed significantly increased sensitivity to PCN (Table S8), which suggested that FgGCN5 is likely the primary target of PCN. Moreover, we added new data demonstrating that PCN can directly bind with FgGcn5 (new data in Figure 4g) and inhibits acetyltransferase activity (new data in Figure 4f) in this new submission.

- The activity assay in figure 4 (mislabeled f. rather than g.) clearly shows that PCN inhibits GCN5, but this does not show that it is the effect of PCN on GCN5 that blocks histone acetylation. Therefore the statement on p 13 “collectively, our evidences indicate that PCN specifically targeted the histone acetyltransferase” is an over statement.

Re: In the revised manuscript, we now provide new evidence suggesting that PCN is able to directly bind to FgGcn5, and subsequently inhibit the acetyltransferase activity. We purified the FgGcn5 protein, tested the inhibitory activity of PCN towards HAT of FgGcn5, and determined the binding affinity between PCN and FgGCN5 using the SPR (Surface Plasmon Resonance). Results clearly indicated that PCN can inhibit the acetyltransferase activity of FgGcn5 *in vitro* (new data in Figure 4f) and that PCN strongly binds to FgGcn5 with a $K_D=71.37$ nM (new data in Figure 4g). Furthermore, we predicted the binding poses of PCN in the HAT of FgGcn5 by using the multiple ligand simultaneous docking (MLSD) program, and found that molecules of PCN were able to bind within the FgGcn5 pocket (new data in Fig. S8, Video S1-3). We rewrote these sentences based on the reviewer’s comments and our new data.

- On line 362 the authors state that fig. S5 shows that deletion mutants of SAGA did not alter the sensitivity towards the antifungal drugs. However the mutants do look attenuated. Is there a mistake?

Re: Deletion mutants of SAGA did not significantly alter the sensitivity towards the antifungal drugs tested. We updated the images and added the statistical analysis data (new data in Figure S9).

- In the results section part 5 it is unclear how any conclusion can be made on the role of the SAGA complex on pathogenicity if the SAGA complex is important for growth.

Re: We believe that the attenuated virulence of the mutants of key components in the SAGA complex on the wheat spikes is mainly caused by:
1. Abolishment of DON biosynthesis. The deoxynivalenol (DON) biosynthesis was totally blocked in the mutants. DON is a critical virulence factor and plays a significant role in the spread of fungus within a spike. The deletion mutants of DON biosynthesis genes (for examples $\Delta tri5$, $\Delta tri6$ and $\Delta tri10$) were significantly reduced the virulence and largely restricted to the inoculated spikelet, [Molecular Plant Pathology, 2006, 7(6):449–461, Molecular Microbiology, 2009, 72(2):354–367].

2. Loss of penetration ability on wheat tissues. The FgGpmk1 mitogen-activated protein kinase cascade and plant cell wall degrading (CWD) enzymes are critical for penetration structure formation on host tissues in *Fusarium graminearum* [Current Genetics, 2003, 43: 87–95; Molecular Plant Pathology, 2015, 16 (1):1–13; New Phytologist, 2015, 206: 315–328; BMC Genomics 2013, 14:274]. Here, we found that the expression of selected genes in FgGpmk1 cascade and CWD encoding genes was significantly decreased in the SAGA mutants (new data, Figure S11). All tested mutants were not able to penetrate the cellophane sheet, similar to the $\Delta FgGpmk1$ mutant (new data, Figure S10a). More importantly, the SEM examination showed that the hyphae of the wild-type PH-1 and complemented strains formed penetration structures on wheat spikelets, but such penetration structures were not observed from the mutants (new data, Figure 5f and Figure S10b). Collectively, these data suggest that the SAGA complex plays an essential role in the pathogenicity by regulating the DON biosynthesis and penetration ability in this fungus.

In addition, we don't think mycelial growth rate is highly related to pathogenicity in *F. graminearum*. The rate of radial growth of $\Delta FgPCN$ and other mutants of SAGA components was reduced to 40-63% in comparison with that of the wild type on the PDA medium, but they totally abolished the

virulence, even on the inoculated spikelets. Whereas, the mutants with similar growth defects were still able to infect the host and cause typical Fusarium head blight symptoms, such as $\Delta FgTIP41$ [New Phytologist, 2014, 203: 219–232] and $\Delta Fgrho3$ [Fungal Genetics and Biology, 2013, 61:90–99]. Collectively, the abolishment of DON biosynthesis and penetration defects were mainly responsible for the non-pathogenic characteristics of the SAGA mutants on wheat, where growth defect of mutants might also play a minor role.

- In general the paper needs to be edited both for content, mainly figure legends, and for English grammar which is lacking throughout.

Re: The language was thoroughly revised by native English speakers and through professional editing service through Nature Research Editing Service (Order ID: 1HLPN8MP, Gold Editing) in this new submission. The Nature Research Editing Service Certification was also submitted for reviewers.

Reviewer #2 (Remarks to the Author):

This is a very interesting, comprehensive and novel study. The initial approach used to find a novel biocontrol strain is very traditional. But then the researchers undertake a series of both 'brute force' and elegant experiments to define the mechanism conferring the inhibitory effects of the Pseudomonas strain ZJU60 against *F. graminearum*. Overall, the manuscript is well presented, well written, laid out and the figures and tables are high quality. This said, there are still several major and more minor concerns, which the authors need to address.

Major concerns

The formation of biofilms and pellicles

Abstract Line 20-21 states 'ZJU60 forms biofilms on the hyphae of *F. graminearum*',. This reviewer struggled to find evidence in the manuscript to support this strong statement The evidence in the results section is as follows Lines 222 – 225 '...observed with scan electron microscopy. After 5 days of co-culture, a large number of ZJU60 cells formed extensive biofilm-like

structure on the surface of fungal mycelium and caused the shrivelled mycelium (Fig. 3d)'. There must be biochemical markers for biofilm formation. The authors cannot rely just on SEM observations on deformed mycelium and the presence of bacteria. Or the authors need to drop entirely the concept of a biofilm. Similarly, the detection of pellicle and biofilm formation in another experiment.

Re: We thank the reviewer for the insightful suggestions. The bacterium ZJU60 cells form extensive fibers and matrix on the surface of fungal mycelium. Secreted matrix is a typical and key feature of bacterial biofilms,

For the biochemical marker of biofilm: exopolysaccharides are a key biofilm matrix component of many bacteria, as they contribute to the overall biofilm architecture. Genome-wide analysis showed a *psl*(polysaccharide locus) homologous gene cluster for exopolysaccharide biosynthesis in ZJU60 (Fig.S4). *psl* encodes a mannose-rich fiber-like structural polysaccharides of biofilm matrix in *P. aeruginosa* (*J Bacteriol* 186, 4457-4465, 2004). Therefore, in the revised manuscript, the biofilms formed by ZJU60 and Δ phzA-H were also quantified by immunoblotting with anti-Psl serum [a gift from Prof. Luyan Ma, the corresponding author of the *Environmental Microbiology*, 2013, 15 (8):2238–2253]. Consistent with the crystal violet staining result, the Psl production in the mutant was dramatically reduced, in comparison with that of wild type ZJU60 in both tested media (new data, Fig.3i in this revised manuscript).

Line 243 – the word 'pellicles' needs to be defined and a suitable reference provided

Re: A pellicle is a biofilm that assembles at the air–liquid interface of a standing liquid culture. It is a type of biofilm structures described in several model bacterial systems (e.g. *Bacillus subtilis*, *Bacillus cereus*, *Pseudomonas aeruginosa*). As suggested, we added a reference in this new submission.

Line 243 – 245 states 'ZJU60 cells formed thin floating pellicles at the air-liquid interface (Fig. 3h, right-hand panels). Meanwhile, cells also formed submerged biofilms in the WA and YEPD media, especially the WA medium (Fig. 3h, top panels). There is no biochemical evidence for either floating pellicles or submerged biofilm formation. These are key points that need to be addressed via additional experimentation.

Re: As suggested, the biofilm formation was quantified by immunoblotting with anti-Psl serum in this new submission. The new data are shown in Figure 3i.

And in another experiment

Line 255-256 In addition, ZJU60 was also form biofilm on the surface of wheat tissues. Again no biochemical evidence for biofilm formation.

Re: We changed the statement to "form a biofilm-like macro colony with fibers on the surface of wheat tissues".

Lines 757-758 Submerged biofilms were quantified by crystal violet staining as follows. Is there a reference for this method of quantification of submerged biofilms? If not how do the authors know that only biofilms are detected by crystal violet staining. Crystal violet is a very general stain used for multiple purposes.

Re: Crystal violet staining is commonly to semi-quantify the biofilms in the field. CV will stain cells that are tightly attached to the bottom surface of the well (after repeated wash) in a submerged biofilm. As suggested, the original reference [George A. O'Toole, Roberto Kolter, Molecular Microbiology, 1998, 28(3):449-61] was added.

Finally, in the discussion, there is no mention of biofilm formation which tends to suggest that the authors to not themselves think this is a very strong part of their study.

Re: Thanks for pointing out this. We do think the observed biofilm-like structures by ZJU60 are significant and worth emphasizing. The discussion on biofilm formation is now added in this revised manuscript.

The microbiome

Lines 125-126 – “It was implied that wheat head might recruit antagonistic bacteria in microbiome for defending infection by the fungus.” – This is a very strong statement for the minimal evidences shown to this point. Therefore, this point should be stated as a hypothesis, not implied based on the data shown so far.

Re: We agree. We rephrased the text based on the reviewer's suggestion.

PCN induction and the eventual model

Lines 217 -221. ‘We extracted and quantified the PCN production in the ZJU60 culture alone and co-culture with *F. graminearum* using LC-MS. As shown in Fig. 3c, the relative PCN production was increased about 50% in co-culture than the ZJU60 culture alone. These data suggested that physical interaction between ZJU60 and *F. graminearum* stimulated the production of the antifungal compound PCN’. This is an exceptionally interesting result. But it is also a very counterintuitive one. This result suggests that the presence of

wild-type *F. graminearum* hyphae leads to the strong induction of a compound by ZJU60 which will have a strong detrimental effect on fungal hyphae health. There would be very strong evolutionary pressure within *F. graminearum* to mutate and not to be an inducer of PCN. This point needs to be considered in the manuscript discussion. This PCN induction data also strongly impacts on the working model given in Figure 6 and has not yet been considered.

Re: Thank you for the very insightful thought. As suggested, we added new discussion about the PCN induction during the bacterial-fungal interactions, and accordingly revised our working model. The phenomenon of phenazine induction during the bacterial-fungal interaction has been reported in the *Pseudomonas aeruginosa-Candida albicans* interaction system (PLoS Pathog 10(10): e1004480, 2014). We also added this reference in this new submission.

Fg mutant analysis

An impressive number of single gene deletion Fg mutants were tested. Some have previously been published, others have not. In lines 296-297 the following are stated ATP-binding cassettes (60 genes), transcription factors (105 genes), and epigenetic modification related proteins (72 genes). But no other information is given on exactly the Fg locus ID evaluated. For completeness an additional supplementary file needs to be included giving the IDs of this additional set of 237 genes. This is valuable negative information that can be used in bioinformatics/ network analyses and also indicates to the global fungal community that these mutant strains are available for both within and cross species comparisons.

Re: We now provided the Fg locus IDs of the 237 tested mutants as Support information data 3 in this new submission.

DON mycotoxin induction

Lines 401 – 404 states 'In addition, DON is synthesized in the specific cellular compartmentalization toxosome in *F. graminearum*, which is mainly formed by trichothecene biosynthesis enzymes encoding by TRI genes. The experiment under evaluation is a wheat head time course involves predominantly extensive in plant colonisation. The two papers cited reveal that the toxosome structure is occurring at the immediate initial post penetration phase or in vitro. There is no evidence to link toxosome formation to any of the later stages of the infection process in the peer reviewed literature. This text in the manuscript will need to be substantially revised to provide the necessary explanation of the DON results provided.

Re: In our very recent publication, we found that the Tri1-GFP labeled

toxosome was formed during the infection process on the host tissues [Figure 1G left-hand panel, PLoS Pathogens, 2018, 14(1):e1006827]. We added this reference in this submission. Meanwhile, the SAGA mutants did not form toxosome under the same condition (new data shown in Figure 5e in the revised manuscript).

Lines 405-406 'we examined toxosome formation by measurement of Tri1-GFP, a key DON biosynthetic enzyme' Exploring the induction of Tri 1, which does not reside in the main TRI biosynthetic cluster on Chromosome 2 is a strange choice to explore DON induction . The authors need to give the rational and cite a reference(s) for why this gene choice is suitable / optimal to explore DON induction.

Re: Although the locus of *TRI1* doesn't reside physically in the main *TRI-5* cluster, *TRI1* is located in the *TRI1-TRI16* cluster; and Tri1 protein plays an essential role in the intermediate step by catalyzing the calonectrin to 7, 8-dihydroxy calonectrin in the DON biosynthesis process. Therefore, we chose Tri1 as a marker for toxosome formation. The following pathway shows the trichothecene biosynthesis from farnesyl pyrophosphate to DON (Biosci. Biotechnol. Biochem., 71 (9), 2105–2123, 2007; Molecular Microbiology, 72(2):354–367, 2009):

-And as suggested, we added the references for that.

Lines 418 – 421 states PCN at either EC50 or EC90 significantly suppressed Tri1-GFP expression, disrupted the toxosome formation, and reduced the DON production in the wild type strain, while carbendazim enhanced the DON biosynthesis (Fig. 5b, c, d). Is the carbendazim result confirming what has already been reported in the literature, or is this the first evidence that another fungicide triggers DON induction? Please clarify for the reader.

Re: Currently, we are still working on the molecular mechanism of carbendazim in triggering DON production although our recent publication has shown this phenomenon [Figure 1, PLoS Pathogens, 2018, 14(1):e1006827]. When we previously submitted the manuscript to this journal, the other manuscript was still under revision by PLoS Pathogens. We now added the new publication as a reference in this revised manuscript.

Comparing mutant phenotypes

Lines 515-517 Due to the fact that DON is responsible for virulence of *F. graminearum*, all SAGA complex mutants had a complete loss of pathogenicity in planta (Fig. 5a). But this argument is somewhat flawed because neither the Tri1 or Tri5 mutants exhibit such a severe reduction in pathogenicity towards wheat heads at the SAGA mutants. This comparative loss of disease causing ability between mutants needs to be further discussed.

Re: We agree with reviewer that SAGA complex mutants exhibited more severe virulence reduction *in planta* than the *TRI* gene mutants. Δ Tri1 or Δ Tri5 and other *TRI* gene mutants were still able to infect the inoculated spikelet, but without extension. In this study, all SAGA complex mutants completely lost their virulence *in planta*, even on the inoculated spikelets. In addition to the DON reduction, these mutants also completely abolished their penetration ability on wheat tissues. The FgGpmk1 mitogen-activated protein kinase cascade and plant cell wall degrading (CWD) enzymes are critical for penetration structure formation on host tissues in *Fusarium graminearum* [Current Genetics, 2003, 43: 87–95; Molecular Plant Pathology, 2015, 16 (1):1–13; New Phytologist, 2015, 206: 315–328; BMC Genomics 2013, 14:274]. Here, we found that the expression of selected genes in FgGpmk1 cascade and CWD genes was significantly decreased in the SAGA mutants (new data, Figure S11). All tested mutants were not able to penetrate the cellophane sheet, similar to the Δ FgGpmk1 mutant (new data, Figure S10a). More importantly, the scanning electron microscopy examination showed that the hyphae of the wild type PH-1 and complemented strains formed penetration structures on wheat spikelets, but such penetration structures were not observed from the mutants (new data, Figure 5f and Figure S10b). Collectively, these data suggest that the SAGA complex plays an essential role in the pathogenicity by regulating the DON biosynthesis and penetration ability in this fungus.

Figures – major points of concern

Figure 1a – The image of bacteria colonies with green crystals is in very poor quality

Re: Thanks for pointing it out. The bacterial colonies with green crystals were re-imaged and updated in this new submission.

Figure 1 – d. The data needs to be presented as absolute values and not as shown comparative data. The reader has no idea of the actual disease pressure in these two field trial years. This will be very important for comparative analyses in the future across treatment types and years.

Re: As suggested, we changed the disease control efficacy (%) with disease index (%). New results are included in the new Figure 1d.

Figure 2 Panel c. There is an excessive amount of aerial mycelium extending from two or the three wheat ears. This is highly unusual when the heads remain attached to the plants and are photographed immediately after detachments. Either the plants were re-placed in a high humidity box for 24 hr prior to the image capture or the heads were detached and placed in high humidity for a period of time before image capture. Please clarify exactly how this image was obtained. This normal types of fusarium infected wheat heads are shown in Figure 5 panel a.

Re: We agree with the reviewer. Since that the wheat heads after treatment were placed in a high humidity growth chamber, the disease symptom shown in Figure 2c was not a typical *F. graminearum* infected wheat heads. To clarify that, we added experiment details in the text as follows: Biocontrol efficacy of ZJU60 against FHB in growth chamber was determined using wheat cultivar Jimai 22 (susceptible to FHB). Briefly, the detached mid-anthesis stage wheat heads were sprayed with a cell suspension of biocontrol strain ZJU60 (10^8 CFU/mL) by a hand-held atomizer. Fungicide phenamacril (2 mg/mL) and sterile water only served as positive and negative controls, respectively. After 6 h of prevention treatments, each treated wheat heads were sprayed with 3 ml conidial suspension of *F. graminearum* strain PH-1 at a concentration of 10^4 conidia per mL with 0.05% Tween 20. The inoculated wheat heads were incubated in the growth chamber. After 7 days of incubation in the growth chamber with high humidity; the infected wheat heads were imaged.

Figure 3 Panel a. Very poor quality needs to be replaced with far better images.

Re: As suggested, we replaced the image.

Figure 4 – Figures ‘e’ and ‘f’ does not match what is in the text and the indication for figure panel ‘g’ is missing.

Re: Thanks for pointing that out. We fixed the error.

Lines 183-187 – “In contrast, the growth inhibitory activities of the mutants that had disrupted biosynthetic operons for hydrogen cyanide, pyoverdine and achromobactin, respectively, were not significantly reduced, when compared to that of the wild type ZJU60 (Fig. 2a)” – Fig. 2a does not show any data related to this statement. It would be good to have in the supplementary material the inhibitory activity of other mutants.

Re: As suggested, we added this result as Figure S1.

Discussion

Phenazines are well known determinants of biological control of soil-borne plant pathogens by various strains of fluorescent *Pseudomonas* spp (Doi 10.1046/j.1469-8137.2003.00686) and have also been tested in the past against *Fusariumoxysporum* (doi: 10.1016/j.micres.2010.06.001). Therefore, phenazines should be discussed in this wider context.

Re: Thanks for the suggestion. We now added discussion about the role of phenazines in biological control in this new submission.

Other changes required

Additional missing references and / or poor explanations and / or incorrect information

Lines 93-95 – “Unfortunately, fungicide-resistant *F. graminearum* strains have been detected in the field after long-term intensive application of the fungicides.” - missing reference.

Re: We added the reference.

Lines 95-96 states ‘Moreover, several fungicides were able to stimulate the mycotoxin production’. A more balanced sentence with additional references is required here because some fungicides do not appear to directly induce DON mycotoxin accumulation whereas others do.

Re: We revised this sentence (lines 95-98) based on the reviewer’s suggestion and added new references.

Missing technical information

Line 136 – Was the strain ZJU60 recovered from healthy or infected wheat heads?

Re: ZJU60 was recovered from infected wheat heads. We added this information in the revised submission.

Lines 143-144 – “Similar to the fungicide phenamacril, the treatment of ZJU60 by foliar spray...” – In this sentence or in the introduction, it would worthy to mention that phenamacril is a fungicide widely used to control FHB in China

Re: Thanks for the suggestion, we now added this information.

Lines 543 – 544 Healthy and natural *F. graminearum* infected wheat heads (Cultivar: Yannong 19) were collected from Anhui province. Is the cultivar Yannong semi-resistant or fully susceptible to FHB?

Re: Yannong 19 is a highly susceptible cultivar to FHB.

Line 560-561 – which stages? How many samples were from healthy and infected?

Re: Wheat head samples were collected at the 50% and 100% flowering stages, respectively, from 4 different fields for cultivable bacteria isolation using the dilution plating method. At each stage and in each field, five individual spots were identified by the five-spot sampling method, and in each spot, 10 healthy and 10 infected wheat heads with 50% disease index were collected. Therefore, a total of 80 independently pooled wheat head samples were collected for cultivable bacterial isolation in this study. We added this information in the new submission.

Lines 574 -576 For ZJU60, its antagonistic activities towards 7 other plant pathogenic fungi including *Alternaria alternata*, *Botrytis cinerea*, *Fusarium oxysporum*, *Fusarium verticillioides*, *Magnaporthe grisea*, *Penicillium italicum* and *Sclerotinia sclerotiorum* were also examined. All these strains need references and / or full details if this is their 1st publication. Please provide as a supplementary table.

Re: All tested plant pathogenic fungi were used in previous studies. We added the names of strains in this new submission.

Line 605 antiSMASH – missing reference

Re: We added the reference.

Ln 659-661: “The primers used to amplify the flanking sequences for each gene are listed in Table S1.” This table is currently missing. Instead Table S1 is “Average relative abundances of the dominant genera in the wheat head microbiome “

Re: We corrected the mistake. We now added the list of primers as the support information data 5 in this submission.

Line 674 Recombination plasmids – what selectable marker was used for the Fg transformations, this is not obvious from the way this paragraph is constructed.

Re: Geneticin was used for the selection marker after transformation. We added the information in the methods.

Line 719 tested strains were grown in TBI medium. What is the full composition of TBI media?

Re: We added a reference for the TBI medium. Liquid TBI medium contained

(per liter) 30 g of sucrose, 1 g of KH_2PO_4 , 0.5 g of $\text{MgSO}_4 \cdot 7\text{H}_2\text{O}$, 0.5 g of KCl, 10 mg of $\text{FeSO}_4 \cdot 7\text{H}_2\text{O}$, 800 mg of putrescine, and 200 μl of trace element solution (5 g of citric acid, 5 g of $\text{ZnSO}_4 \cdot 7\text{H}_2\text{O}$, 0.25 g of $\text{CuSO}_4 \cdot 5\text{H}_2\text{O}$, 50 mg of $\text{MnSO}_4 \cdot \text{H}_2\text{O}$, 50 mg of H_3BO_3 , and 50 mg of $\text{NaMoO}_4 \cdot 2\text{H}_2\text{O}$ per 100 ml).

Line 729-732 – The reference Liu et al., 2015 describe the pathogenicity assay using Fg spores. If they used mycelia, how mycelia concentration was measured and how much was used?

Re: Thanks for pointing this out. We deleted the inappropriate reference. Fresh mycelia (0.1 mg) were used as inoculums for each spikelet.

Minor points

Lines 48 and 52. The acronym SGAG is incorrect and should be changed to SAGA

Re: We fixed it

Line 65. 'compositions' a better word choice is required

Re: The "compositions" was replaced by "two major groups of microbiome".

Line 114 - In the rhizosphere, increasing evidences have plants are able to recruit.... Sentence structure does not seem to be right, rewrite

Re: We fixed that.

Lines 124 – if more than one species of Pseudomonas, it should be Pseudomonas spp.

Re: We fixed that.

Lines 129-130. 'Therefore, we are toward to use a simple cultivable bacterial-F. graminearum interaction system...'. Poor English , sentence needs to be rewritten.

Re: We revised this sentence.

Line 393 Due to F. graminearum is a plant pathogen,... Poor English, sentence needs to be rewritten.

Re: We revised this sentence.

Line 574 – 'alternat' should be 'alternata'

Re: We fixed that.

Line 575 – 'grisea' should be in italic

Re: We fixed that.

Reviewer #3 (Remarks to the Author):

Comment to the manuscript entitled "Epigenetic regulation mediated by a member of the wheat head microbiome reduces virulence and growth of a major wheat fungal pathogen"

The manuscript describes a novel mechanism for Fusarium disease control by Pseudomonas. The manuscript provides a comprehensive dataset, and the

rationale behind the experiments is well described. However, in multiple occasions, the description and the presentation of the data should be improved substantially to meet the quality of the journal. See further the comments below.

The introduction is poor. The English writing needs improvement. The authors refer to several examples of bacterial fungal interactions, but many more have been described. The authors state several times that the molecular mechanism behind bacterial-fungal interactions are unknown, but they have been described for several systems. Either the authors are unaware, or perhaps trying to say something else that is not coming across. Since the authors present a new type of mechanism of how a potential biocontrol bacterium can control a pathogenic fungus, the authors should include the different mechanisms that have been described so far for bacteria-fungal interactions to control fungal plant pathogens (for example antimicrobial compounds and volatiles, mycophagy, induction of plant resistance, etc) so they can better emphasise the novelty of their finding.

Re: Thanks for the suggestion. Most of the described bacterial-fungal direct interactions to control fungal plant pathogens were achieved via antimicrobial compounds secreted by antagonistic bacteria in agroecosystems. Although many antifungal compounds have been identified, their direct targets in fungi and the mode of action were not well understood. That is why we stated in the manuscript that "the underlying mechanisms are largely unknown". In this study, we identified the target of the antifungal compound phenazine-1-carboxamine in *F. graminearum*, and revealed a novel mechanism at the epigenetic level for the BFI. In the revised manuscript, we have toned down the statement and added discussion about known modes of action between bacterial-fungal interactions to control fungal disease in agriculture ecology in the introduction.

In the results section, several statements are made that are incorrect, see detailed comments below.

An impressive array of analyses have been performed, including microbial community analysis of wheat head, bacterial isolations, Fusarium growth inhibiting assays, fatty acid analysis, biocontrol assays in climate chamber and field, phylogeny of concatenated housekeeping genes, genome sequencing, mutagenesis of *Pseudomonas*, LC-MS, SEM and TEM, colonization and biofilm formation of *Pseudomonas*, mutagenesis of multiple pathways in *Fusarium* (ROS-associated, SAGA, acetyltransferase), mutagenesis of *Botrytis*, and *Magnaporthe*, mutant library screenings of *Fusarium* and *Saccharomyces*, immunoblotting, histone acetyltransferase inhibition, toxosome

formation.

Even though the paper describes this large amount of data, the presentation of the data is sometimes unclear, and the result sections should be substantially revised. Figures should be carefully checked for spelling errors. Also figure legend and/or methods section should be transparent and detailed. Often statistical analyses, what error bars indicate etc are not described.

Detailed comments

Re: We have made substantial changes in every section in the revised manuscript and reached out to Nature Research Editing Service to thoroughly edit the manuscript. We hope that the writing of the revised manuscript is significantly improved.

In various sections the English spelling is incorrect or unclear, a few are included below.

L114: incomplete sentence

Re: We fix that.

L117: 16S sequences cannot give information about the protective status. Rephrase.

Re: We revised this sentence.

L119: mention how many OTUs were obtained and how many families/genera.

Re: The sequences were grouped into 482 and 600 different OTUs (Operational Taxonomic Units) from healthy and infected wheat head samples, respectively, and a total of 38 genera were identified. We added this information in the revised manuscript.

L125: I don't think it is just to say that this is 'recruitment'. It could also be a response of the community because the wheat head environment is changing because of the infection and might leak other nutrients. I suggest rephrasing

Re: As suggested, we rephrased this sentence (lines 128-130).

L129: unclear sentence

Re: We fixed that.

L133: infected or uninfected? Did the author focus on a specific family of bacteria based on the microbiome analysis? Did they use specific media?

Re: We recovered cultivable bacterial isolates from both healthy and infected wheat heads. Five different media were used to isolate bacteria from the

above samples in this study. We tried to recover cultivable bacterial species as many as possible, not limited to a specific family.

L137: on top of its colony.

Re: We changed it.

L143: 'material and methods' is labelled as 'method'

Re: We fixed that.

L153: Please indicate whether these are high scores for the identification. Not all readers are aware if these values are good or not.

Re: As suggested, we added the following information in the methods.

[Interpretation Guidelines in Microbial Identification System: Use the following guidelines when interpreting the Similarity Index. Samples with a similarity of 0.500 or higher with a separation of 0.100 between the first and second choice are considered good library comparisons. If the Similarity Index is between 0.300 and 0.500 and well separated from the second choice (>0.100 separation), it may be a good match, but an atypical strain. Values lower than 0.300 suggest that we do not have the species in the database, but the software will indicate the most closely related species.

L175-176: how are these genes distinctly different? This cannot be seen in Table S5. It would be helpful to add identity percentages with known HCN, PCN, PCA and achromobactin genes.

Re: We revised this sentence. In the previous manuscript, we wanted to emphasize that these biocontrol agents were proposed to secret different antifungal compounds, but not the variation of encoding gene sequences for those antifungal compounds.

L187: fig 2a is not referring to the data described in the sentence.

Re: Thanks for pointing it out. It should be Figure 2b. We fixed that.

L210: how do authors know that the green substance is PCN?

Re: The green pigment is an important feature for subspecies identification of *Pseudomonas chlororaphis*. In addition, the pure PCN compound is in green color. We revised the text and added the reference.

L221: what about PCA production?

Re: The production of PCA, as the precursor of PCN, was also induced but mildly at about 10%.

L225: it is not proven that the biofilm formation of the bacteria is causing the shrivelled mycelium

Re: We agree and we revised the sentence. The shrivelled mycelium was caused by the PCN, not the biofilm itself.

L229: Fig S2a is not showing the Δ phzA-H vs *F. graminearum* co-culture. Spelling *F. graminearum* incorrect.

Re: Fig.S2a (Fig. S3a in this new submission) showed the mycelium of *F. graminearum* without treatment. The Δ phzA-H vs *F. graminearum* co-culture data were shown in Fig. 3e. We fixed the spelling error.

L235: unclear how total culturable cells and specific bacteria have been obtained. Typing error y-axis 3g.

Re: We described the details in the methods, and fixed the spelling error.

L251: Fig 3i showing SEM not present.

Re: Fig. 3i was shown in the middle panels. We reorganized the order of images in Fig. 3.

L263: what is meant with “chemical genomic analysis”?

Re: Chemical genomics is a reverse-genetics approach that uses genome-wide mutant collections to gain functional insight into the modes-of-actions of chemical compounds. We added more information and references for this approach in the methods in this new submission.

L270: what is meant with “interacting gene mutants”?

Re: Using the chemical genomic analysis, we identified a total of 90 mutants with potentially altered sensitivity to PCN. To further explore whether those potential genes were indeed involved in PCN sensitivity, 90 mutants and the mutants of their interactors at the protein level were retrieved from deletion mutant library. These genes encoding the proteins that directly interact with these 90 gene encoding proteins were identified based on data in the yeast genome data (www.yeastgenome.org).

L309: It can be debated whether Spt7 is really homologues with only 20% identity as described in L268. Was the Spt7 mutant of *S. cerevisiae* screened in the mutants described in L263-270? If so, could the authors explain why this mutant did not show a change in susceptibility to PCN?

Re: The FgPCN shares 27% identity with Spt7 in yeast and both of the genes encode Bromo_SPT7_like domain. The two genes are defined as orthologs in

the EggNOG database (acc. ENOG4103J17), and reciprocal blast search in each proteome indicates that they are the best hit in the other genome. Based on the evidence above, we believe that FgPCN is an ortholog of Spt7.

-Our results suggest that PCN targets the histone acetyltransferase SAGA complex, and inhibits the acetyltransferase activity in *F. graminearum*. By contrast, PCN did not influence the growth of SPT7 mutant of yeast. This variation may be due the difference of SAGA architecture between the species, which would be an interesting topic for future investigation.

L367: when comparing the protein sequences of Spt7 and FgPCN, can the authors find a reason for the clear separation between fungi and yeast homologs? Are there any domains different?

Re:As suggested, we analyzed the domain(s) of Spt7 and FgPcn homologs using the SMART online software (<http://smart.embl-heidelberg.de/>). Results showed that both the yeast SPT7 and FgPcn homologs had a bromodomain [Bromodomains are found in a variety of mammalian, invertebrate and fungal chromatin-associated proteins, which can interact with acetylated lysine, and might be involved in protein-protein interactions and may play a role in assembly or activity of multi-component complexes involved in transcriptional activation, (PUBMED:9175470, PUBMED:7580139)]. Moreover, FgPCN or its homologs in filamentous fungi harbors an additional Bromo_TP domain in their C-terminus. Bromodomain Transcription factors and PHD domain is found in many chromatin-associated proteins. The domain differentiation might be one of reasons for the clear separation. (Schematic diagram of domains in SPT7 and FgPCN was followed.)

L393: unclear English

Re: We revised the sentence.

L401: unclear what is meant with 'a spike'

Re: A spike means a tassel.

L452: Is Listeria the only pathogenic bacteria described for this phenomenon?

Describe a few.

Re: Up to date, at least 4 pathogenic bacteria have been reported to affect the chromatin structure and transcriptional program of host cells by altering histone acetylation, including *Listeria monocytogenes*, *Mycobacterium tuberculosis*, *Helicobacter pylori* and *Anaplasma phagocytophilum*. We revised the sentence and added references.

L486: be consistent in naming of H3K27 etc, it is different here compared to L334 and in Fig 4.

Re: We fixed that.

L516: Since it is apparently easy to construct mutants in Fusarium: has a TRI-mutant lost its virulence?

Re: YES, a *TRI*-mutant lost its virulence. The deletion mutants of DON biosynthesis genes (such as $\Delta tri5$, $\Delta tri6$ and $\Delta tri10$) were reduced in virulence dramatically and restricted to the inoculated spikelet [Molecular Plant Pathology, 2006, 7(6):449–461, Molecular Microbiology, 2009, 72(2):354–367]. We also tested the virulence of $\Delta tri1$ and $\Delta tri4$ in our experiments. The results are in agreement with the finding from $\Delta tri5$, $\Delta tri6$ and $\Delta tri10$. Here are the images.

L558: please describe the methods used for analysing the sequencing data. The reference provided is not describing 16S amplicon data analysis.

Re: Thanks for pointing it out. We described the methods in the revised manuscript.

Fig 1f: what each ring in the genome is representing should be described in figure legend.

Re: We updated the figure of the genome and the legend.

Fig 2: order of panels is not logic. (2d above b/c). In 2g: include pictures of the control (0 ug/ml).

Re: We reorganized the panels and the control treatment was also added in Figure 2g in this new submission.

Table S4: include the accession numbers of the other *P. chlororaphis* isolates.

Re: The accession numbers were added in Table S4.

Fig S4b: present data in graph by measuring hyphal diameter.

Re: As suggested, the inhibition of mycelial growth rate (%) by each fungicide towards fungal strains was statistically analyzed (new data in Figure S9).

Reviewers' comments:

Reviewer #1 (Remarks to the Author):

In the revised version of "epigenetic regulation mediated by a member of the wheat head microbiome reduces virulence and growth of a major wheat fungal pathogen", the authors have added data, modified the text and in general improved their manuscript. The manuscript demonstrates well that PCN inhibits *F. graminearum* growth and controls head blight. In addition the manuscript has been improved and demonstrates well the interaction and the effect of PCN on the SAGA complex. In general this study represents a tremendous amount of work and presents interesting and novel findings. However there are 3 points that remain to be addressed:

- The revised manuscript does not provide any more data on the epigenetics topic. The data shows nicely that there is a difference in modified histone levels, but this is not sufficient to claim that there is an epigenetic regulation. However, this is not a serious problem as the work is interesting without it. Nonetheless, it is recommended to change the title to remove the word "epigenetics", which is currently misleading.
- The phenotypes of the mutants presented in this study has not been sufficiently addressed. Indeed, all mutants of the SAGA complex show a growth defect even without PCN. At least this defect should be addressed in the discussion. In addition, if PCN only affects GCN5 activity, one would expect the phenotype of the GCN5 mutant (without PCN) to mimic that of the WT+PCN. However, in figure 4 it is clear that this is not the case and the GCN5 mutant seems more defective in growth than the WT+PCN. Please address this point.
- In general the data is not clearly quantified and statistically analyzed. For example, figures 4, 5, S9 do not mention any statistical analysis for the data besides p values. How are these calculated? Similarly in the text, line 395 "showed significantly increased". How is it determined that it is significant? In table S8 many other mutants are affected by more than 50%, this seems like a high percentage. Please address these points.

Reviewer #2 (Remarks to the Author):

The authors have addressed most comments and included additional data. This reviewer is OK with these changes.

However, in the discussion a one-paragraph summary is missing at the beginning of this section which summarises the major findings. Perhaps this could be inserted at line 497. Instead at the moment there is a one paragraph summary of the phenazine literature (ln 475) (as requested by one of the reviewers). So overall the article is not an easy read.

Reviewer #3 (Remarks to the Author):

The manuscript has improved substantially and became much more clear and easier to follow. Specifically, the addition of figure 4f and 4g provide evidence for direct effect of PCN on the SAGA complex.

Some minor comments:

L62. Rephrase : However, there is an increasing need for deciphering microbe-microbe interactions in

the plant microbiome and the functions of interactions that drive the dynamic microbial community at the molecular level.

L137: culturable instead of cultivable

L157: delete of

L237-244: the difference between 3f and 3g is not clear, not in the results section nor in figure legend. Both seem to describe the populations of mycelium associated bacteria. Based on the methods I presume that 3f describes the data of vortexed co-cultures, so including planktonic cells and those attached to the mycelium; while in 3g the data is representing only the bacterial cells that were attached to mycelium and excluding the planktonic cells.

L307 the tested should be then tested

L278-294: Since the assay was done in *S. cerevisiae* and the 4 *Fusarium* mutants have no effect, I would shorten this section even more or even remove from manuscript since it is not adding anything. The mutant screening of *Fusarium* is more important.

L463 How is the penetration of cellophane and expression of penetration-related genes when exposed to PCN? This can apparently be easily measured with qRT-PCR.

L551 please add a discussion on why less biofilm formation is observed in the PCN mutant. Is PCN regulating the biofilm formation, how and why?

Figure legend 1e: explain what DON contamination means.

Figure legend 3f/3g: describe better, because now it is not clear.

Figure legend 3h: write WA and YEPD medium in full.

Response to referees

Reviewers' comments:

Reviewer #1 (Remarks to the Author):

In the revised version of “epigenetic regulation mediated by a member of the wheat head microbiome reduces virulence and growth of a major wheat fungal pathogen”, the authors have added data, modified the text and in general improved their manuscript. The manuscript demonstrates well that PCN inhibits *F. graminearum* growth and controls head blight. In addition the manuscript has been improved and demonstrates well the interaction and the effect of PCN on the SAGA complex. In general this study represents a tremendous amount of work and presents interesting and novel findings. However there are 3 points that remain to be addressed:

- The revised manuscript does not provide any more data on the epigenetics topic. The data shows nicely that there is a difference in modified histone levels, but this is not sufficient to claim that there is an epigenetic regulation. However, this is not a serious problem as the work is interesting without it. Nonetheless, it is recommended to change the title to remove the word “epigenetics”, which is currently misleading.

Re: Thanks for suggestion. We changed the title to "Suppression of histone acetylation mediated by a member of wheat microbiome reduces phytopathogenic fungal virulence".

- The phenotypes of the mutants presented in this study have not been sufficiently addressed. Indeed, all mutants of the SAGA complex show a growth defect even without PCN. At least this defect should be addressed in the discussion. In addition, if PCN only affects GCN5 activity, one would expect the phenotype of the GCN5 mutant (without PCN) to mimic that of the WT+PCN. However, in figure 4 it is clear that this is not the case and the GCN5 mutant seems more defective in growth than the WT+PCN. Please address this point.

Re: Thank you very much for the great point. We agree with the reviewer that all mutants of the SAGA complex show a growth defect even without PCN. That is because the SAGA-mediated histone modification is a critical regulatory for growth in fungi.

- SAGA mediates histone acetylation of gene promoters to enhance transcriptional activation by recruiting the pre-initiation complex and facilitates elongation by deubiquitinating monoubiquitinated histone H2B (H2B-Ub) and

acetylating histones within the coding region to promote histone eviction (Anders M. Näär et al., *Annu Rev Biochem*, 2001; J.A. Daniel et al., *J Biol Chem*, 2004; Chhabi K. Govind et al., *Mol Cell*, 2007). Removal of monoubiquitin from Lys¹²³ of H2B is required for recruitment of the Ctk1 kinase and subsequent Ser2 phosphorylation of the RNA polymerase II C-terminal domain (A. Wyce *et al.*, *Mol Cell*, 2007). In addition to promoting gene activation and transcription elongation, SAGA also contributes the export of the nascent mRNA through the nuclear pore complex (P. Pascual-Garcia *et al.*, *Genes Dev*, 2008). These multiple functions of SAGA contribute together to regulate the expression of target genes, and play essential roles in developmental processes and environmental fitness in eukaryotes. For example, loss of dGcn5 is lethal because of the lack of metamorphosis in *Drosophila* (Carre C *et al.*, *Mol Cell Biol* 2005). In a similar manner, Gcn5 is required for normal growth and development in mice and Arabidopsis (Xu W et al., *Nat Genet* 2000; Konstantinos E. et al., *Plant cell*, 2003). The biological function of SAGA complex in regulating growth is also conserved in fungi. In *S. cerevisiae*, the SAGA complex is required for the normal transcription of approximately 10% of genes (e.g., ribosome biogenesis, translation, amino acid metabolism), that are predominantly associated with growth and stress response (Huisinga, K.L. et al., *Mol. Cell*, 2004; Ling Cai et al., *Mol Cell*, 2011). In filamentous fungi, the SAGA complex is also critical for mycelial growth. For example, disruption of the SAGA complex results in growth defects in *A. nidulans* and *F. fujikuroi* (Canovas D, *et al.*, *Genetics*, 2014; Rosler SM *et al.*, *Mol Microbiol*, 2016). Consistent with these previous reports from other fungi, we found that disruption of the SAGA complex led to significantly reduced mycelial growth (Fig. 4a, Supplementary Fig. 6) in *F. graminearum*. In general, the SAGA complex regulates transcription of many genes, and some of them may be associated with vegetative growth. However, the exact genes remain unidentified yet in *F. graminearum*. We discuss the growth defect of mutants in the discussion part.

-The growth inhibition of PCN towards the wild-type is largely depended on the dosage. In the Figure 4a, the EC₅₀ of PCN (15 µg/mL) was supplemented in the agar plates. When the final concentration of PCN is increased, the mycelial growth of WT+PCN will be more defective than that of ΔGCN5 mutant, for example, EC₉₀ (25 µg/mL) in Figure 2g. Our results have shown that the histone acetyltransferase GCN5 is a major target of PCN. In addition, PCN may also affect other pathways, which causes different morphologies between the GCN5 mutant and the wild type treated with PCN. Such phenomenon is very common in pharmacology. For example, the fungicide carbendazim

targets both beta-tubulin 1 and 2, but the beta-tubulin 2 is the major target in *F. graminearum*.

- In general the data is not clearly quantified and statistically analyzed. For example, figures 4, 5, S9 do not mention any statistical analysis for the data besides *p* values. How are these calculated? Similarly in the text, line 395 “showed significantly increased”. How is it determined that it is significant? In table S8 many other mutants are affected by more than 50%, this seems like a high percentage. Please address these points.

Re: Thanks for pointing it out. In this manuscript, all statistical analyses were performed using SAS software. Data are presented as mean ± standard deviation (SD). Differences between two groups were analyzed by Student's *t* test. Multiple comparisons were analyzed by one-way analysis of variance (ANOVA) followed by least significant difference (LSD) multiple-range test. The normality of data was tested using the Kolmogorov-Smirnov test ($P < 0.05$), and the equality of error variances was tested by Levene's test ($P < 0.05$). In the case of nonnormality and/or unequal variances, data were transformed before ANOVAs. We added this information in the methods in this version.

-For the supplementary table 8: To test the sensitivity of putative acetyltransferase gene mutants, the EC₅₀ concentration of PCN (15 µg/mL) was used. The growth inhibition rate of the wild-type PH-1 was (55.56±5.31) % (Mean±SD) under this condition. In the last submission, we did not show the growth inhibition rate of the wild-type PH-1, and the standard deviation (SD) of the PCN susceptibility of mutants. We added these data in this new submission (new supplementary table 8), and the data were analyzed by LSD test at $P = 0.01$. Result indicated that only ΔFgGCN5 showed significantly increased sensitivity to PCN (new supplementary table 8).

Reviewer #2 (Remarks to the Author):

The authors have addressed most comments and included additional data. This reviewer is OK with these changes.

However, in the discussion a one-paragraph summary is missing at the beginning of this section which summarises the major findings. Perhaps this could be inserted at line 497. Instead at the moment there is a one paragraph summary of the phenazine literature (ln 475) (as requested by one of the reviewers). So overall the article is not an easy read.

Re: Thanks for suggestion. We added one paragraph summary in this submission, and revised the manuscript to make it easy read according to reviewers' suggestions.

Reviewer #3 (Remarks to the Author):

The manuscript has improved substantially and became much more clear and easier to follow. Specifically, the addition of figure 4f and 4g provide evidence for direct effect of PCN on the SAGA complex.

Some minor comments:

L62. Rephrase: However, there is an increasing need for deciphering microbe-microbe interactions in the plant microbiome and the functions of interactions that drive the dynamic microbial community at the molecular level.

Re: We changed the statement as the reviewer's suggested.

L137: culturable instead of cultivable

Re: Thanks. We changed the "cultivable" to "culturable" in this submission.

L157: delete of

Re: We fixed that.

L237-244: the difference between 3f and 3g is not clear, not in the results section nor in figure legend. Both seem to describe the populations of mycelium associated bacteria. Based on the methods I presume that 3f describes the data of vortexed co-cultures, so including planktonic cells and those attached to the mycelium; while in 3g the data is representing only the bacterial cells that were attached to mycelium and excluding the planktonic cells.

Re: We fixed the label in the figure legends.

L307 the tested should be then tested

Re: We fixed that.

L278-294: Since the assay was done in *S. cerevisiae* and the 4 *Fusarium* mutants have no effect, I would shorten this section even more or even remove from manuscript since it is not adding anything. The mutant screening of *Fusarium* is more important.

Re: Thanks for suggestion. These data support the finding that PCN has different inhibitory mechanisms against the budding yeast and *F. graminearum*. Thus, we did not remove the whole paragraph, but as suggested by the reviewer, we shortened this section in this new submission.

L463 How is the penetration of cellophanes and expression of penetration-related genes when exposed to PCN? This can apparently be easily measured with qRT-PCR.

Re: As suggested, we conducted the penetration of cellophane assay and

expression of penetration-related genes in the wild-type PH-1, when it was exposed to PCN. In consistent with the observation from Δ FgGCN5, the penetration capability of PH-1 into cellophanes was dramatically reduced in the present of PCN (15 μ g/mL), and almost abolished when the wild type strain treated with a concentration of PCN (20 μ g/mL) (new supplementary Fig.10b). Meanwhile, the mRNA expression levels of penetration-related genes were decreased in the PCN treatment (new supplementary Fig.11).

L551 please add a discussion on why less biofilm formation is observed in the PCN mutant. Is PCN regulating the biofilm formation, how and why?

Re: Thanks for the suggestion. We added the discussion on the link of biofilm formation and the production of phenazines in this submission.

-Previous studies have shown that the production of phenazines is tight linked with the process of biofilm formation in phenazine producing *Pseudomonas* spp. (S. Chincholkar and L. Thomashow (eds.), *Microbial Phenazines*, 2013). For example, the mutations in regulatory and structural phenazine genes caused deficiency in biofilm formation of *P. chlororaphis* and *P. aeruginosa*. Addition of phenazines from cell-free culture supernatants or as purified phenazine was able to restore biofilm formation of mutations to the wild-type levels in both microtiter plate, flow cell assays and and plant roots (V.S.R.K. Maddula *et al.*, *Microbial ecol*, 2006; Wang Yun *et al.*, *J bacteriol*, 2011). In consistent with these previous reports, we found that the deletion of PCN biosynthetic gene cluster (phzA-H) caused deficiency in biofilm formation of ZJU60 in microtiter plates and on the mycelia (Figs. 3h-j).

- Phenazines could contribute to biofilm development in various ways: (1) Phenazines may regulate biofilm formation partly through bacterial surface migration. Swarming motility was significantly higher in Δ phz1/2 than that of the wild-type *P. aeruginosa* PA14, and then higher motility caused thinner biofilm (Mikkel Klausen *et al.*, *Mol Microbiol*, 2003; Itzel Ramos *et al.*, *Res Microbiol*, 2010). (2) Phenazines promote extracellular DNA (eDNA) release and stimulate biofilm formation in *Pseudomonas* spp. eDNA is one of the major factors dictating the progression of biofilm formation. It has been shown that pyocyan interacted with molecular oxygen, resulting in production of reactive oxygen species which can lead to cellular injury and release of DNA (Theerthankar Das *et al.*, *Microbial biofilms*, 2016). (3) Phenazines promote bacterial biofilm development via ferrous iron acquisition. Iron is an important signal for *P. aeruginosa*, affecting twitching motility and thereby biofilm attachment. Phenazine-1-carboxylic acid (PCA) can promote biofilm development by reducing Fe (III) to ferrous iron [Fe (II)], thereby making iron

more bioavailable (Wang Yun *et al.*, *J bacteriol*, 2011). (4) Phenazines may also serve as signals, triggering the expression of other important factors in biofilm development (V.S.R.K. Maddula *et al.*, *Microbial ecol*, 2006). However, phenazines influencing the dynamics of biofilm formation are often affected by environmental conditions and appear to be species-specific (Parejko JA *et al.*, *Appl Environ Microbiol*, 2013; Melissa K. LeTourneau *et al.*, *Environ Microbiol*, 2018). The exact mechanism on the role of PCN in regulating ZJU60 biofilm formation is worth to be determined in the future.

Bacterial biofilms serve to increase adhesion on biotic and abiotic surfaces, resistance to antibiotics, heat, and other environmental stresses, and likely more competitive for limited available resources. However, the consequences of life in a biofilm include the limitations in oxygen, iron, and other nutrients. In *Pseudomonas* spp., phenazines play a role in survival within the aggregate bacterial community. For example, phenazines act as electron acceptors for the reoxidation of NADH to balance the intracellular redox state of cells, thereby promote survival of cells in deeper anoxic layers of the biofilm (Lars E. P. Dietrich *et al.*, *Science*, 2008; Koley D *et al.*, *Proc Natl Acad Sci USA*, 2011; Dietrich LE, *et al.*, *J bacteriol*, 2013). Meanwhile, phenazines can reduce Fe^{3+} to Fe^{2+} and increase the bioavailability of iron for the biofilm formation of *Pseudomonas* spp. (Wang Yun *et al.*, *J bacteriol*, 2011).

Taken together, the positive feedback between phenazine production and biofilm formation increases the ecological fitness in Phz^+ *Pseudomonas* spp. In our case, our data shown that PCN was critical for ZJU60 biofilm formation and the PCN production was increased during the biofilm formation on the mycelium (Figure 3). These results suggested that the feedback loop of PCN production and biofilm formation may play an important role during the bacterial-fungal interaction in environmental conditions.

Figure legend 1e: explain what DON contamination means.

Re: DON contamination means the production of DON in the wheat kernels. We changed the figure legend to "Efficiencies of ZJU60 and phenamacril in controlling DON production in field trials".

Figure legend 3f/3g: describe better, because now it is not clear.

Re: We fixed the label in the figure legends.

Figure legend 3h: write WA and YEPD medium in full.

Re: We fixed that.